# Poly2Vec: Polymorphic Fourier-Based Encoding of Geospatial Objects for GeoAI Applications

**Maria Despoina Siampou** [* 1]  **Jialiang Li** [* 2]  **John Krumm** [1]  **Cyrus Shahabi** [1]  **Hua Lu** [3]

## Abstract

Encoding geospatial objects is fundamental for geospatial artificial intelligence (GeoAI) applications, which leverage machine learning (ML) models to analyze spatial information. Common approaches transform each object into known formats, like image and text, for compatibility with ML models. However, this process often discards crucial spatial information, such as the object's position relative to the entire space, reducing downstream task effectiveness. Alternative encoding methods that preserve some spatial properties are often devised for specific data objects (e.g., point encoders), making them unsuitable for tasks that involve different data types (i.e., points, polylines, and polygons). To this end, we propose POLY2VEC, a polymorphic Fourier-based encoding approach that unifies the representation of geospatial objects, while preserving the essential spatial properties. POLY2VEC incorporates a learned fusion module that adaptively integrates the magnitude and phase of the Fourier transform for different tasks and geometries. We evaluate POLY2VEC on five diverse tasks, organized into two categories. The first empirically demonstrates that POLY2VEC consistently outperforms object-specific baselines in preserving three key spatial relationships: topology, direction, and distance. The second shows that integrating POLY2VEC into a state-of-the-art GeoAI workflow improves the performance in two popular tasks: population prediction and land use inference.

---

[*]Equal contribution. Jialiang Li's work was conducted while the author was a visiting student at USC's InfoLab. [1]Department of Computer Science, University of Southern California, Los Angeles, USA [2]Department of People and Technology, Roskilde University, Denmark [3]Department of Computer Science, Aalborg University (Copenhagen campus), Denmark. Correspondence to: Maria Despoina Siampou <siampou@usc.edu>.

*Proceedings of the 42$^{nd}$ International Conference on Machine Learning*, Vancouver, Canada. PMLR 267, 2025. Copyright 2025 by the author(s).

## 1. Introduction

The increasing availability of geospatial data from sources such as satellites, ground-based sensors, and crowdsourced platforms like OpenStreetMap (OSM)[1] (Lee & Kang, 2015; Jokar Arsanjani et al., 2015; Basiri et al., 2019), combined with the recent advancements in machine learning (ML) (Vaswani, 2017; Bommasani et al., 2021), has fueled significant progress in geospatial artificial intelligence (GeoAI) (Smith, 1984; Couclelis, 1986; Janowicz et al., 2020; Gao et al., 2023). GeoAI leverages ML models to analyze geospatial objects, such as points of interest (POIs), building footprints, and vehicle trajectories, thereby extracting valuable insights that enable a variety of decision-making applications, including transportation network optimization (Li et al., 2018b; Mirowski et al., 2018), urban planning (Zhang et al., 2021; Wu et al., 2022), energy management (Sun et al., 2020), and improved emergency response strategies (Kyrkou et al., 2022), to name a few.

A fundamental step in GeoAI pipelines is the transformation of geospatial data into latent representations that can be easily processed by ML models, a step formally known as *encoding*. A common approach to encoding converts coordinate-based geospatial data into formats compatible with established feature extraction models. Although effective for specific tasks, this conversion often discards crucial spatial information, significantly limiting the generalizability of these models. For example, building footprints are frequently rasterized into images and processed with vision-based models for urban prediction tasks (Li et al., 2023; Balsebre et al., 2024). While this approach captures object shapes, it neglects important spatial relationships, such as the relative positioning and alignment of objects within the area. Similarly, POIs that are represented as text, by using attributes like category as input to language-based models, capture semantic relationships but omit precise spatial locations (Huang et al., 2022). As a result, these approaches are application-specific and struggle to generalize across tasks that require a deeper understanding of spatial relationships.

To address the aforementioned limitations, spatially explicit encoding techniques have been proposed. These methods

---

[1]https://openstreetmap.org

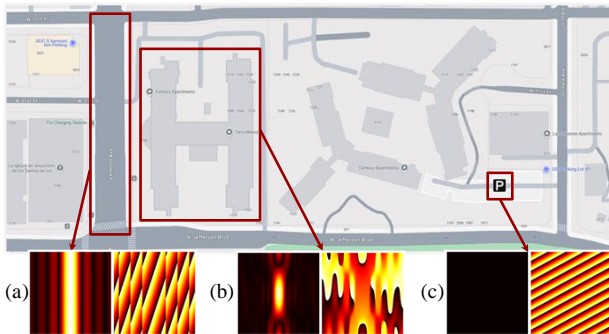

Figure 1: Visualization of the Fourier transform magnitude and phase of (a) road segment, (b) building, and (c) POI.

preserve crucial spatial properties, while remaining compatible with downstream ML models. For instance, THEORY (Mai et al., 2020) encodes the absolute positions of POIs using sinusoidal functions with varying frequencies. Xu et al. (2018) directly encodes the coordinates of trajectories using multi-layer perceptons and feeds it to a GRU, capturing their sequential nature. For polygons, NUFT-SPEC (Mai et al., 2023) maps geometries into the spectral domain, effectively preserving key polygon properties such as topology awareness. However, the design of these methods inherently limits their applicability, as they only capture the properties of the specific geospatial object they are devised for. This restricts their generalizability in tasks involving mixed geospatial object types, such as land use classification, where integrating points (e.g., POIs) and polygons (e.g., buildings) requires simultaneously preserving their spatial properties as well as relationships between them.

In this work, we introduce POLY2VEC[2], a polymorphic encoding framework that unifies the representation of 2D geospatial objects, including points, polylines, and polygons. At its core, POLY2VEC leverages the Fourier transform to encode essential spatial properties, transforming the input geometries[3] into the frequency domain. Given that this transformation results in complex-valued features, the magnitude and phase components are extracted. As shown in Figure 1, these components complement each other: the magnitude reflects spatial extent, being uniform for points with no shape and varying for polygons and polylines, while the phase highlights directionality, such as the alignment of a polyline. To combine these components into a single representation, POLY2VEC incorporates a learned fusion module that adaptively balances their contributions based on the task and geometry type, producing a real-valued geometry embedding that ensures compatibility with ML models.

---

[3]We refer to geometries and geospatial objects interchangeably.

We formally define four key properties, shape preservation, direction preservation, distance preservation, and task flexibility, as essential criteria for evaluating the effectiveness of geometry encoding. These properties ensure the produced embeddings accurately capture the essential geometry characteristics while remaining versatile across different tasks. To demonstrate that POLY2VEC preserves these properties, we conduct a two-part evaluation. First, we evaluate POLY2VEC on spatial reasoning tasks, showing that it outperforms the state-of-the-art specialized baselines by up to 17% for topological classification, 26% for directional classification, and 75% for distance estimation. Second, we show that integrating POLY2VEC into a state-of-the-art GeoAI workflow reduces prediction error by 14% and 5% in population prediction and land use inference.

In summary, our contributions are:
• We introduce POLY2VEC, the first encoding framework that unifies the representation of various 2D geometries.
• We propose a 2D continuous Fourier transform-based encoding approach to capture crucial spatial properties, including shape, distance, and direction.
• We design a learned fusion strategy to adaptively combine Fourier magnitude and phase for diverse objects and tasks.
• Our experiments show that POLY2VEC preserves crucial geometry encoding properties, demonstrating its versatility in handling diverse geospatial objects, and task-flexibility when integrated into state-of-the-art GeoAI pipelines.

## 2. Preliminaries

### 2.1. Problem Formulation

**Definition 1** (Geospatial Object). A geospatial object $g \in \mathbb{R}^2$ is represented by an array $P \in \mathbb{R}^{N \times 2}$, where each row is a point $(x, y)$, and $N$ is the total number of points. The type of geometry (e.g., point, polyline, or polygon) is determined by the organization and relationships among these points.

**Polymorphic Encoding of Geospatial Objects.** Given a dataset of geospatial objects $G = \{g\} \in \mathbb{R}^{N \times 2}$, the goal is to define an encoding function $e_\theta(g) : \mathbb{R}^{N \times 2} \rightarrow \mathbb{R}^d$, parameterized by $\theta$, that maps each geometry $g$ to a $d$-dimensional vector $\mathbf{v}$, termed as geometry embedding. The embedding dimension $d$ remains constant across different geometry types, making $e_\theta$ polymorphic. The encoding is intended to capture the following key properties.

**Property 1** (Shape Preservation). For any geometry $g \in G$, its embedding $\mathbf{v}$, should capture its structural characteristics: shape and boundary for polygons, length for polylines, and the lack of spatial extent for points.

**Property 2** (Direction Preservation). For any geometries $g_i, g_j \in G$, $e_\theta$ should ensure their embeddings $\mathbf{v}_i, \mathbf{v}_j$ reflect their relative orientation.

**Property 3** (Distance Preservation). For any geometries

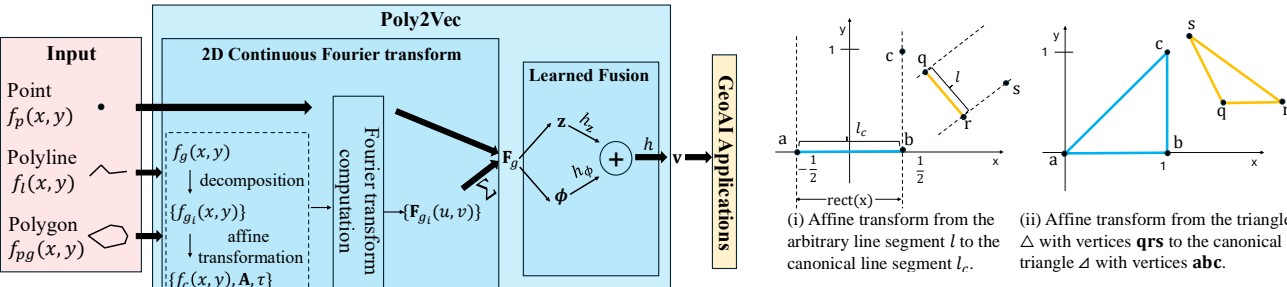

(a) The workflow of POLY2VEC.

(b) Affine transform arbitrary geometry to its corresponding canonical geometry.

Figure 2: Overview of POLY2VEC.

$g_i, g_j \in G$, the similarity of their embeddings $\mathbf{v}_i, \mathbf{v}_j$ should monotonically decrease as their spatial distance $\|g_i - g_j\|$ increases, and vice versa.

**Property 4** (Task Flexibility). The encoder $e_\theta$ should facilitate multiple tasks without requiring modifications.

Properties 1-3 ensure that $\mathbf{v}$ captures all essential spatial information, while Property 4 guarantees flexibility for use across GeoAI models. Section 4 empirically demonstrates that our proposed $e_\theta$ satisfies these properties.

### 2.2. 2D Continuous Fourier Transform Properties

A key component of our encoding approach is the computation of the 2D continuous Fourier transform (CFT) [4]. For a given 2D function $f(x, y)$, its Fourier transform is denoted as $\mathscr{F}\{f(x, y)\} = F(u, v)$[5] and is formally defined as:

$$F(u, v) = \int_{-\infty}^{\infty} \int_{-\infty}^{\infty} f(x, y) e^{-j2\pi(ux+vy)} dx \, dy \quad (1)$$

where $j = \sqrt{-1}$ and $u, v$ are the frequency samples.

We now summarize Fourier transform properties relevant to our approach following (Gaskill, 1978).

**Linearity.** The Fourier transform of a sum of functions denoted as $f_i(x, y)$, is the sum of their corresponding Fourier transforms $F_i(u, v)$:

$$\mathscr{F}\left\{\sum_{i=1}^{n} a_i f_i(x, y)\right\} = \sum_{i=1}^{n} a_i F_i(u, v), \quad a_i \in \mathbb{C} \quad (2)$$

**Affine Transformation.** For an affine-transformed function $f(\mathbf{A}\mathbf{x} + \boldsymbol{\tau})$, where $\mathbf{x} = [x, y]^\top$, its Fourier transform is:

$$\mathscr{F}\{f(\mathbf{A}\mathbf{x}+\boldsymbol{\tau})\} = \frac{1}{|\det(\mathbf{A})|} e^{-j2\pi\boldsymbol{\tau}^\top \mathbf{A}^{-\top}\mathbf{u}} F(\mathbf{A}^{-\top}\mathbf{u}) \quad (3)$$

where $\mathbf{u} = [u, v]^\top$, $\mathbf{A} \in \mathbb{R}^{2\times 2}$ is the affine matrix, and $\boldsymbol{\tau} \in \mathbb{R}^2$ is the translation vector.

[4]We use Fourier Transform and CFT interchangeably.
[5]For compactness, we use $F(u, v)$ to describe the CFT.

**Hermitian Symmetry.** For real-valued functions $f(x, y)$, $F(u, v)$ satisfies $F(u, v) = F^*(-u, -v)$, where $F^*(u, v)$ denotes the complex conjugate.

**Magnitude and Phase.** The Fourier Transform $F(u, v)$ is a complex-valued function composed of a real part, $\text{Re}(F(u, v))$, and an imaginary part, $\text{Im}(F(u, v))$. The magnitude $z(u, v)$ and phase $\phi(u, v)$ are defined as:

$$z(u, v) = \sqrt{\text{Re}(F(u, v))^2 + \text{Im}(F(u, v))^2} \quad (4)$$

$$\phi(u, v) = \text{atan2}(\text{Im}(F(u, v)), \text{Re}(F(u, v))) \quad (5)$$

## 3. Methodology

Figure 2 illustrates our proposed POLY2VEC, which uniformly encodes arbitrary geospatial objects for GeoAI applications. We first describe how the Fourier transform is derived for each geometry type, and then outline the learned fusion module for deriving the final geometry embeddings.

### 3.1. 2D Continuous Fourier Transform of Geometries

#### 3.1.1. FOURIER TRANSFORM OF A POINT

A point $p = (x_p, y_p) \in \mathbb{R}^2$ is modeled as a 2D Dirac delta function, which represents the point as a distribution concentrated entirely at $(x_p, y_p)$, and can be expressed as:

$$f_p(x, y) = \delta(x - x_p, y - y_p) \quad (6)$$

To that extent, the Fourier transform of $f_p(x, y)$ is given by:

$$F_p(u, v) = e^{-j2\pi(x_p u + y_p v)} \quad (7)$$

where $(u, v)$ are the frequency components.

The Fourier transform magnitude for any point is constant, $z_p(u, v) = 1$, while the phase $\phi_p(u, v)$ encodes its location.

As shown in Figure 2, deriving the Fourier transform for polylines and polygons involves additional steps. Polylines are divided into line segments, and polygons are triangulated into non-overlapping triangles. The Fourier transform is

computed for each component by affine transforming them to their canonical geometry, and the linearity property of Eq. (2) is used to compute the Fourier transform of the original geometry[6]. Details for polylines and polygons are specified below, with derivation details in Appendix A.2.

### 3.1.2. FOURIER TRANSFORM OF A POLYLINE

We begin by deriving the Fourier transform of a canonical line segment and then generalize to any arbitrary line segments. Consider the canonical line segment $l_c$, which extends from $\mathbf{a} = (-\frac{1}{2}, 0)$ to $\mathbf{b} = (\frac{1}{2}, 0)$ in $\mathbb{R}^2$, as shown in Figure 2b. Then, $l_c$ can be represented as:

$$f_{l_c}(x, y) = \text{rect}(x)\delta(y) \qquad (8)$$

where $\delta(y)$ represents a Dirac delta function ridge along the $x$-axis, and $\text{rect}(x)$ restricts the ridge to the interval $|x| \leq \frac{1}{2}$.

The Fourier transform of $f_{l_c}(x, y)$ is given by:

$$F_{l_c}(u, v) = \text{sinc}(u) \qquad (9)$$

Now consider an arbitrary line segment $l$ with endpoints $\mathbf{q} = (x_q, y_q)$ and $\mathbf{r} = (x_r, y_r)$. To compute the Fourier transform of $l$, we map it to the canonical line segment $l_c$, using the affine transformation property. To compute this, we first introduce an auxiliary point $\mathbf{c} = (\frac{1}{2}, 1)$ to the structure of $l_c$ so that it is not colinear with $\mathbf{ab}$. This point maps to another auxiliary point $\mathbf{s}$ introduced in the structure of the arbitrary line segment $l$. The auxiliary point $\mathbf{s}$ is defined as $\mathbf{s} = \mathbf{r} + \mathbf{n}$, where $\mathbf{n} = (y_q - y_r, x_r - x_q)^\top$, representing a 90° clockwise rotation of the vector $\mathbf{r} - \mathbf{q}$. Note that the line segments $\mathbf{qr}$ and $\mathbf{rs}$ have the same length.

Given the points $\mathbf{q}, \mathbf{r}, \mathbf{s}$ and $\mathbf{a}, \mathbf{b}, \mathbf{c}$ we then construct the affine transformation matrix $\mathbf{A} = [\mathbf{a} \, \mathbf{b} \, \mathbf{c}][\mathbf{q} \, \mathbf{r} \, \mathbf{s}]^{-1}$. By applying Eq. (3), the Fourier transform of an arbitrary line segment $l$, with endpoints $\mathbf{q}, \mathbf{r}$, is expressed as:

$$F_l(u, v) = \frac{1}{|\det(A)|} e^{-j2\pi\boldsymbol{\tau}^\top \mathbf{A}^{-\top}\mathbf{u}} F_{l_c}(\mathbf{A}^{-\top}\mathbf{u})$$
$$= \|\mathbf{q} - \mathbf{r}\|^2 e^{-j2\pi\left(\frac{\mathbf{q}+\mathbf{r}}{2}\right)\mathbf{u}} \text{sinc}(\mathbf{u}^\top(\mathbf{r} - \mathbf{q})) \quad (10)$$

At $(u, v) = (0, 0)$, the Fourier transform is $F_l(0, 0) = \|\mathbf{q} - \mathbf{r}\|^2$, the squared length of the line segment.

Finally, following Eq. (2), the Fourier transform of an arbitrary polyline $pl$ is computed as:

$$F_{pl}(u, v) = \sum_{i=1}^{T_l} F_{l_i}(u, v) \qquad (11)$$

where $F_{l_i}(u, v)$ is the Fourier transform of the $i$-th line segment and $T_l$ is the total number of line segments.

### 3.1.3. FOURIER TRANSFORM OF A POLYGON

To compute the Fourier transform of a polygon we decompose it into a set of non-overlapping triangles using standard triangulation techniques[7]. We thus begin with the Fourier transform of a canonical isosceles right triangle and then generalize to its computation for arbitrary triangles.

Consider the canonical isosceles right triangle $\triangle_c$ with vertices $\mathbf{a} = (0, 0)$, $\mathbf{b} = (1, 0)$, and $\mathbf{c} = (1, 1)$, represented as:

$$f_{\triangle_c}(x, y) = \begin{cases} 1, & \text{if } 0 \leq x \leq 1 \text{ and } 0 \leq y \leq x, \\ 0, & \text{otherwise.} \end{cases} \qquad (12)$$

The Fourier transform of $f_{\triangle_c}(x, y)$ is then given by[8]:

$$F_{\triangle_c}(u, v) = \int_0^1 \int_0^x e^{-j2\pi(ux+vy)} \, dy \, dx$$
$$= \frac{1}{4\pi^2 uv(u+v)} \Big[ \big((u+v)\cos(2\pi u)$$
$$- u\cos(2\pi(u+v)) - v\big) - j\big((u+v)\sin(2\pi u)$$
$$- u\sin(2\pi(u+v))\big) \Big] \qquad (13)$$

Next, we compute the Fourier Transform of an arbitrary triangle $\Delta$, with vertices $\mathbf{q} = (x_q, y_q)$, $\mathbf{r} = (x_r, y_r)$, and $\mathbf{s} = (x_s, y_s)$, by mapping it to the canonical triangle using the affine transformation property ( Figure 2b). The affine transformation matrix is defined as $\mathbf{A} = [\mathbf{a} \, \mathbf{b} \, \mathbf{c}][\mathbf{q} \, \mathbf{r} \, \mathbf{s}]^{-1}$.

By substituting the vertices of $\Delta$ into $\mathbf{A}$ and applying Eq. (3), the Fourier Transform of the triangle $F_\Delta(u, v)$ can be calculated. In this computation, the determinant of $\mathbf{A}$, $|\det(\mathbf{A})| = \frac{1}{2\alpha}$, where $\alpha$ is the area of the triangle $\Delta$.

Finally, the Fourier transform of an arbitrary polygon $pg$, given the linearity property of Eq. (2), can be computed as:

$$F_{pg}(u, v) = \sum_{i=1}^{T_{pg}} F_{\Delta_i}(u, v) \qquad (14)$$

where $F_{\Delta_i}(u, v)$ is the Fourier transform of the $i$-th triangle, and $T_{pg}$ is the total number of extracted triangles.

Building on the Fourier transform computation described earlier, we can now extract the frequency representation of a given geometry $g$, expressed as a spatial function $f_g(x, y)$ over $\mathbb{R}^2$ as, $\mathbf{F}_g = [\mathbf{F}_1, \mathbf{F}_2, \ldots, \mathbf{F}_W]^\top \in \mathbb{C}^W$, where $W$ is the number of frequency components, and $\mathbf{F}_i = F(u_i, v_i)$ represents the value of the Fourier transform evaluated at the specific frequency coordinates $(u_i, v_i)$.

---

[6]The same methodology can be adopted to compute the CFT of multi-polygons.

[7]We adopt Constraint Delauney triangulation in this paper.

[8]Special cases where $u$, $v$, and $u+v$ approach zero are handled separately, and presented in Appendix A.2.3.

To sample the frequency components, we employ a geometric series sampling strategy (Mai et al., 2023), which balances low and high-frequency components to capture both global and local details. We also experimented with learned frequency sampling but found that the two strategies produced nearly identical results. Detailed comparisons are presented in Appendix A.3.2.

## 3.2. Learned Fusion of Fourier Transform Features

Given that $\mathbf{F}_g$ consists of complex values, we decompose it in two real-valued vectors of the magnitude $\mathbf{z}$ and the phase $\boldsymbol{\phi}$, computed as in Eqs. (4) and (5), respectively. This transformation ensures the representation is compatible with downstream ML models, which typically operate on real-valued inputs. Furthermore, the magnitude $\mathbf{z}$ captures the intensity of contributions at different frequencies, reflecting the geometry's size and overall shape, while the phase $\boldsymbol{\phi}$ encodes positional and orientational information of the geometry's features (Zahn & Roskies, 1972).

While the final geometry embedding can be created by simply concatenating $\mathbf{z}$ and $\boldsymbol{\phi}$, their relative importance should vary with the geometry type and the downstream task. For instance, the magnitude of points is always 1, whereas it encodes the shape and size of polygons. Therefore, when encoding points, the phase should contribute more than the magnitude in the representation. To this end, POLY2VEC adaptively learns the importance of magnitude and phase through two separate transformations $\mathbf{z}^*=\mathbf{h}_z(\mathbf{z})$ and $\boldsymbol{\phi}^*=\mathbf{h}_\phi(\boldsymbol{\phi})$, where $\mathbf{h}_z: \mathbb{R}^W \to \mathbb{R}^W$ and $\mathbf{h}_\phi: \mathbb{R}^W \to \mathbb{R}^W$ are separate MLPs for $\mathbf{z}$ and $\phi$ respectively.

Finally, the transformed vectors $\mathbf{z}^* \in \mathbb{R}^W$ and $\boldsymbol{\phi}^* \in \mathbb{R}^W$ are concatenated and passed through a final MLP $\mathbf{h}: \mathbb{R}^W \to \mathbb{R}^d$ to produce the final geometry embedding $\mathbf{v} = \mathbf{h}([\mathbf{z}^*; \boldsymbol{\phi}^*]) \in \mathbb{R}^d$, which can be inputted to any machine learning model $M$, such that $M(\mathbf{v}) \to \mathbf{y}$, where $\mathbf{y}$ represents task-specific outputs. We will empirically verify that $\mathbf{v}$ preserves the key properties in Section 4.

## 4. Experiments

In this section, we conduct experiments to evaluate the effectiveness of POLY2VEC across four key research questions:
**[RQ1]** Does POLY2VEC effectively preserve the critical geometric properties of shape, direction, and distance?
**[RQ2]** How does POLY2VEC perform in comparison to baseline encoding methods tailored for specific object types?
**[RQ3]** Can integrating POLY2VEC into existing workflows lead to improvements in their performance?
**[RQ4]** Does learned fusion boost POLY2VEC performance?

### 4.1. Spatial Reasoning Tasks

This section addresses **RQ1** and **RQ2**, empirically evaluating POLY2VEC's ability to preserve the properties of Section 2.1, against specialized baselines. We frame these evaluations as spatial reasoning tasks, which are fundamental to broader applications like geospatial question answering (GeoQA), and require accurate spatial understanding (Punjani et al., 2018; Papamichalopoulos et al., 2024).

**Datasets.** We evaluate our approach on two OpenStreetMap (OSM) datasets from Singapore and New York. Each dataset includes three types of geospatial objects: POIs (represented as points), main roads (as polylines), and buildings (as polygons). All input geometry coordinates are normalized to the range $[-1, 1] \times [-1, 1]$. Additional dataset statistics and preprocessing details are provided in Appendix A.4.

**Baselines.** We include three categories of baselines: **point encoders:** (i) DIRECT, directly utilizing coordinates (Chu et al., 2019), (ii) TILE, a discretization method (Berg et al., 2014), (iii) WRAP, a coordinate wrapping mechanism (Mac Aodha et al., 2019), (iv) GRID, inspired by position encoding (Mai et al., 2020), and (v) THEORY, a multiscale encoder (Mai et al., 2020). All point encoders are extended to other geometries handling them as sequences of points, following Rao et al. (2020); Xu et al. (2018). **polyline encoder:** (i) T2VEC a classic trajectory encoder (Li et al., 2018a). **polygon encoders:** (i) RESNET1D (Mai et al., 2023) and (ii) NUFTSPEC (Mai et al., 2023). More details on baselines can be found in Appendix A.5.

#### 4.1.1. TOPOLOGICAL RELATIONSHIP CLASSIFICATION

This task classifies topological relationships defined by the DE-9IM model (Clementini et al., 1993) for geospatial object pairs. We present the supported relationships in Table 3.

**Settings.** The geometry embeddings of each pair are concatenated, passed through a 2-layer MLP with $N_C$ output units, , corresponding to the number of relationships. We adopt cross-entropy loss for optimization. Performance is measured by accuracy, precision, recall, and F1-score. Accuracy results are presented in Table 1, while the remaining metrics are presented in Appendix A.10.

**Results.** From Table 1, we observe that POLY2VEC consistently outperforms all baselines across all experiments. Unlike specialized encoders that excel only for specific pairs, POLY2VEC's performance is consistent across all geometries, highlighting its versatility and generalization capabilities. The second-best performing models vary by geometry type, with T2VEC for polylines and NUFTSPEC for polygons. This shows that simply extending point encoders to handle all geospatial objects is not adequate, as it fails to preserve characteristics like the object's shape and position, leading to decreased performance. Finally, all mod-

Table 1: Model accuracy on topological relationship classification. **Best** and second best are highlighted.

| Methods | Singapore | | | | | New York | | | | |
|---|---|---|---|---|---|---|---|---|---|---|
| | point-polyline | point-polygon | polyline-polyline | polyline-polygon | polygon-polygon | point-polyline | point-polygon | polyline-polyline | polyline-polygon | polygon-polygon |
| RESNET1D | - | - | - | - | $0.457_{0.017}$ | - | - | - | - | $0.452_{0.033}$ |
| NUFTSPEC | - | - | - | - | $0.602_{0.009}$ | - | - | - | - | $0.585_{0.008}$ |
| T2VEC | - | - | $0.728_{0.023}$ | - | - | - | - | $0.807_{0.121}$ | - | - |
| DIRECT | $0.823_{0.013}$ | $0.843_{0.005}$ | $0.733_{0.007}$ | $0.368_{0.010}$ | $0.357_{0.018}$ | $0.846_{0.011}$ | $0.909_{0.018}$ | $0.745_{0.008}$ | $0.495_{0.009}$ | $0.446_{0.023}$ |
| TILE | $0.790_{0.021}$ | $0.700_{0.010}$ | $0.505_{0.005}$ | $0.459_{0.013}$ | $0.411_{0.009}$ | $0.659_{0.013}$ | $0.783_{0.007}$ | $0.502_{0.009}$ | $0.494_{0.038}$ | $0.405_{0.005}$ |
| WRAP | $0.886_{0.003}$ | $0.880_{0.008}$ | $0.716_{0.011}$ | $0.476_{0.010}$ | $0.349_{0.004}$ | $0.886_{0.006}$ | $0.880_{0.017}$ | $0.733_{0.009}$ | $0.550_{0.011}$ | $0.381_{0.007}$ |
| GRID | $0.846_{0.004}$ | $0.844_{0.004}$ | $0.697_{0.031}$ | $0.458_{0.004}$ | $0.335_{0.012}$ | $0.822_{0.039}$ | $0.891_{0.004}$ | $0.739_{0.009}$ | $0.516_{0.008}$ | $0.381_{0.031}$ |
| THEORY | $0.892_{0.003}$ | $0.900_{0.005}$ | $0.719_{0.008}$ | $0.450_{0.010}$ | $0.461_{0.041}$ | $0.897_{0.008}$ | $0.909_{0.008}$ | $0.734_{0.008}$ | $0.591_{0.006}$ | $0.455_{0.041}$ |
| POLY2VEC | **$0.955_{0.007}$** | **$0.949_{0.002}$** | **$0.812_{0.010}$** | **$0.509_{0.008}$** | **$0.702_{0.006}$** | **$0.953_{0.003}$** | **$0.980_{0.002}$** | **$0.830_{0.004}$** | **$0.641_{0.062}$** | **$0.684_{0.008}$** |

Table 2: Model accuracy on directional relationship classification. **Best** and second best are highlighted.

| Methods | Singapore | | | | | | New York | | | | | |
|---|---|---|---|---|---|---|---|---|---|---|---|---|
| | point-point | point-polyline | point-polygon | polyline-polyline | polyline-polygon | polygon-polygon | point-point | point-polyline | point-polygon | polyline-polyline | polyline-polygon | polygon-polygon |
| RESNET1D | - | - | - | - | - | $0.819_{0.010}$ | - | - | - | - | - | $0.747_{0.010}$ |
| NUFTSPEC | - | - | - | - | - | $0.807_{0.008}$ | - | - | - | - | - | $0.698_{0.017}$ |
| T2VEC | - | - | - | $0.268_{0.075}$ | - | - | - | - | - | $0.249_{0.032}$ | - | - |
| DIRECT | $0.880_{0.006}$ | $0.841_{0.007}$ | $0.844_{0.006}$ | $0.820_{0.002}$ | $0.830_{0.005}$ | $0.752_{0.017}$ | $0.877_{0.004}$ | $0.766_{0.005}$ | $0.836_{0.008}$ | $0.653_{0.007}$ | $0.784_{0.004}$ | $0.694_{0.004}$ |
| TILE | $0.253_{0.001}$ | $0.268_{0.002}$ | $0.273_{0.008}$ | $0.326_{0.010}$ | $0.454_{0.001}$ | $0.394_{0.003}$ | $0.245_{0.009}$ | $0.258_{0.005}$ | $0.316_{0.005}$ | $0.217_{0.001}$ | $0.466_{0.001}$ | $0.349_{0.012}$ |
| WRAP | $0.861_{0.018}$ | $0.804_{0.009}$ | $0.803_{0.004}$ | $0.781_{0.002}$ | $0.831_{0.002}$ | $0.778_{0.001}$ | $0.809_{0.004}$ | $0.669_{0.001}$ | $0.749_{0.018}$ | $0.596_{0.019}$ | $0.772_{0.002}$ | $0.602_{0.006}$ |
| GRID | $0.882_{0.007}$ | $0.728_{0.007}$ | $0.771_{0.003}$ | $0.699_{0.001}$ | $0.641_{0.016}$ | $0.534_{0.138}$ | $0.868_{0.002}$ | $0.590_{0.003}$ | $0.646_{0.049}$ | $0.438_{0.004}$ | $0.752_{0.001}$ | $0.485_{0.079}$ |
| THEORY | $0.912_{0.014}$ | $0.867_{0.009}$ | $0.858_{0.004}$ | $0.834_{0.012}$ | $0.860_{0.006}$ | $0.735_{0.044}$ | $0.892_{0.017}$ | $0.760_{0.007}$ | $0.826_{0.008}$ | $0.684_{0.009}$ | $0.775_{0.005}$ | $0.555_{0.012}$ |
| POLY2VEC | **$0.932_{0.006}$** | **$0.935_{0.032}$** | **$0.925_{0.002}$** | **$0.906_{0.010}$** | **$0.907_{0.007}$** | **$0.833_{0.006}$** | **$0.909_{0.012}$** | **$0.891_{0.004}$** | **$0.883_{0.013}$** | **$0.863_{0.007}$** | **$0.876_{0.009}$** | **$0.785_{0.003}$** |

Table 3: Topological relationships of geometry pairs.

| Geometry Pair | Topological Relationships (a relationship b) |
|---|---|
| point-polyline | disjoint, intersects |
| point-polygon | disjoint, contains |
| polyline-polyline | disjoint, intersects |
| polyline-polygon | disjoint, touches, intersects, within |
| polygon-polygon | disjoint, touches, intersects, contains, within, equals |

els perform better when detecting relationship is a binary classification (e.g., point-polyline in Table 3), compared to multi-classification (e.g., polygon-polygon in Table 3). This is expected, as the latter requires capturing fine-grained spatial nuances, posing greater difficulty. In summary, these results emphasize the importance of preserving shape (Property 1), and distance (Property 3) in geometry embeddings. POLY2VEC's ability to do so, along with its unified framework, enables it to consistently outperform baselines.

### 4.1.2. DIRECTIONAL RELATIONSHIP CLASSIFICATION

This task classifies the directional relationships defined by the 16-compass direction model of two geospatial objects .

**Settings.** We follow the same setting as in Section 4.1.1, with $N_c = 16$, and report the same metrics. Accuracy results are in Table 2, with the rest in Appendix A.11.

**Results.** From Table 2, we observe that POLY2VEC consistently outperforms all baselines across all experiments. This demonstrates its ability to effectively preserve the direction (Property 2) among diverse geometry types. While polygon encoders outperform the extended point encoders also in this task, T2VEC underperforms. This is due to T2VEC's strategy of assigning coordinates to grid cells during encoding, which is effective for trajectory-related tasks, but introduces discretization artifacts that affect angular relationships. A similar limitation is observed in the performance of TILE, which also relies on discretizing points into grid cells. In contrast, POLY2VEC encodes geometries holistically, preserving their relative orientation and avoiding these pitfalls.

### 4.1.3. DISTANCE ESTIMATION

This task evaluates whether geometry embeddings preserve pairwise distances (Property 3).

**Settings.** The original distance is estimated by the Euclidean distance of the geometry embeddings. The mean squared

Figure 3: Distance scatter plots of point-polygon pairs on Singapore dataset for different encoders.

error (MSE) is adopted as loss function. We compare the differences between the predicted and original distances in Figure 3 and report the mean absolute error (MAE) in Appendix A.12.

**Results.** Figure 3 depicts that the predicted distances generated by POLY2VEC are closely aligned with the original distances, whereas the predicted distances from other point encoders appear more scattered. This highlights POLY2VEC's superior ability to preserve spatial distance relationships across various geometry types. Methods like DIRECT are overly simplistic, while approaches such as TILE, GRID, and WRAP introduce location distortions through discretization or periodic transformations, affecting the distance preservation. By leveraging the Fourier transform, POLY2VEC effectively captures both the positions and relative spatial relationships of the geometry pairs, enabling it to implicitly encode distance as a core property into its embeddings.

### 4.2. Integration In an End-to-End GeoAI Pipeline

The section addresses **RQ3**, demonstrating the benefits of integrating POLY2VEC into an existing GeoAI workflow.

**Dataset.** We utilize the same dataset as in Section 4.1. The regions for both cities are extracted using the administrative boundaries of Singapore Subzones and NYC Census Tracts.

**Baseline.** We adopt REGIONDCL (Li et al., 2023), an unsupervised urban region representation learning framework that uses buildings and POIs from OSM for **land use inference** (predicting urban functional distributions) and **population prediction** (estimating region population). REGIONDCL encodes building footprints by converting their coordinates into images and extracting features using ResNet18 while using categorical features for POIs. To address the loss of location information, REGIONDCL employs a distance-biased transformer, which introduces a bias in the self-attention mechanism to prioritize closer objects. We provide more details in Appendix A.9.

**Settings.** We evaluate three variants: (1) REGIONDCL, the original framework, (2) REGIONDCL w/o distance-bias removes the distance-biased term, and (3) REGIONDCL w/ Poly2Vec removes the distance-biased term and replaces the encoding part of the pipeline with POLY2VEC. The training and evaluation strategies remain unchanged across

the variants following the original work. For land use inference, we report L1-distance, KL-divergence, and cosine similarity metrics. For population prediction, we report MAE, root mean squared error (RMSE), and coefficient of determination ($R^2$).

**Results.** The results for both tasks are presented in Table 4. Removing the distance-bias term from REGIONDCL leads to a noticeable drop in performance, emphasizing the importance of encoding the spatial location and alignment of objects for accurate land use and population predictions. When POLY2VEC is added, the performance improves significantly. This shows that POLY2VEC can adequately capture the shape and orientation of objects, similar to the initial image-based features, while also benefiting from the inclusion of object's location. Overall, POLY2VEC encodes spatial information directly into its embeddings, removing the need for additional mechanisms like the distance-bias term. This improves performance while simplifying the pipeline, showcasing the task flexibility of POLY2VEC and its potential for effective integration into GeoAI workflows.

### 4.3. Ablation Study

This section addresses **RQ4**, highlighting the benefits of the proposed learned fusion module.

**Settings.** We include three variants: (1) w/mag uses only the Fourier transform magnitude, (2) w/phase uses only the phase, and (3) w/concat combines both via concatenation.

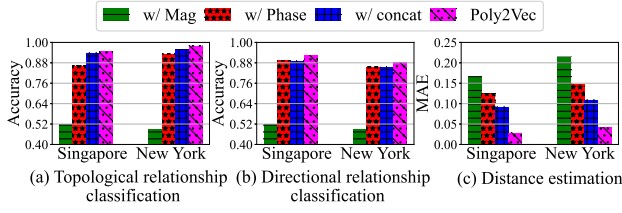

Figure 4: Ablation study for the point-polygon dataset.

**Results.** As shown in Figure 4, among the variants, w/ mag performs the worst across all tasks, particularly in directional relationship classification, as the Fourier transform magnitude primarily captures shape, which is insufficient on its own to address these tasks. In contrast, w/ phase, which

Table 4: Comparison of methods for Land Use Classification and Population Prediction. **Best** values are highlighted.

| | Land Use Classification | | | | | |
|---|---|---|---|---|---|---|
| **Methods** | **Singapore** | | | **New York** | | |
| | L1 $\downarrow$ | KL $\downarrow$ | Cosine $\uparrow$ | L1 $\downarrow$ | KL $\downarrow$ | Cosine $\uparrow$ |
| RegionDCL | $0.498 _{\pm 0.038}$ | $0.294 _{\pm 0.047}$ | $0.879 _{\pm 0.021}$ | $0.418 _{\pm 0.012}$ | $0.229 _{\pm 0.013}$ | $0.912 _{\pm 0.006}$ |
| RegionDCL w/o distance-bias | $0.558 _{\pm 0.043}$ | $0.369 _{\pm 0.067}$ | $0.844 _{\pm 0.023}$ | $0.439 _{\pm 0.012}$ | $0.244 _{\pm 0.012}$ | $0.904 _{\pm 0.005}$ |
| RegionDCL w/ Poly2Vec | $\mathbf{0.484} _{\pm 0.021}$ | $\mathbf{0.278} _{\pm 0.025}$ | $\mathbf{0.881} _{\pm 0.012}$ | $\mathbf{0.397} _{\pm 0.010}$ | $\mathbf{0.212} _{\pm 0.011}$ | $\mathbf{0.923} _{\pm 0.007}$ |
| | Population Prediction | | | | | |
| **Methods** | **Singapore** | | | **New York** | | |
| | MAE $\downarrow$ | RMSE $\downarrow$ | $R^2 \uparrow$ | MAE $\downarrow$ | RMSE $\downarrow$ | $R^2 \uparrow$ |
| RegionDCL | $5807.54 _{\pm 522.74}$ | $7942.74 _{\pm 779.44}$ | $0.427 _{\pm 0.108}$ | $5020.20 _{\pm 216.63}$ | $6960.51 _{\pm 282.35}$ | $0.575 _{\pm 0.039}$ |
| RegionDCL w/o distance-bias | $6018.94 _{\pm 641.71}$ | $8214.58 _{\pm 931.11}$ | $0.385 _{\pm 0.087}$ | $5293.04 _{\pm 277.31}$ | $7348.86 _{\pm 374.62}$ | $0.532 _{\pm 0.030}$ |
| RegionDCL w/ Poly2Vec | $\mathbf{4957.58} _{\pm 506.02}$ | $\mathbf{6874.47} _{\pm 851.73}$ | $\mathbf{0.561} _{\pm 0.117}$ | $\mathbf{4602.75} _{\pm 179.66}$ | $\mathbf{6393.38} _{\pm 279.70}$ | $\mathbf{0.621} _{\pm 0.037}$ |

encodes location information, performs better since relative location, here, is more crucial. Combining both through w/ concat shows improvements, highlighting the importance of integrating both shape and location information. In contrast, POLY2VEC outperforms all variants by employing a learned fusion strategy that adaptively balances the contribution of magnitude and phase based on the task and geometry type. Particularly, this strategy benefits POLY2VEC more in tasks such as point-related distance estimation, where points lack spatial extent, and thus magnitude should contribute significantly less than the phase containing location information. More ablation studies are presented in Appendix A.13.

## 5. Related Work

Existing geometry encoding approaches often focus on one shape type, with point encoders receiving the most attention. Direct point encoding methods simply feed raw coordinates into neural networks but fail to capture details of location distributions (Xu et al., 2018; Chu et al., 2019). Discretization methods assign points to predefined grid cells, as seen in approaches leveraging location context for image classification (Tang et al., 2015; Berg et al., 2014), but struggle with fixed resolution and imprecise representations. Sinusoidal methods encode normalized coordinates using sinusoidal functions, such as WRAP, which captures cyclic patterns (Mac Aodha et al., 2019). Extensions like multi-scale encoder (Zhong et al., 2020) introduce multiple sinusoidal scales. THEORY improves this by computing the dot product of coordinates with unit vectors separated by $120°$ (Mai et al., 2020). There are also point encoders that jointly model location and neighborhood features (Qi et al., 2017; Yin et al., 2019; Zhou & Tuzel, 2018).

Unlike points, there are no dedicated approaches for encoding polylines in their generic form. The closest relevant work is trajectory encoding, where trajectories are often rep-

resented as ordered sequences of points. Most approaches rely on discretization. For instance, Li et al. (2018a) uses grid-based encoding, training an RNN on degraded data to infer missing information and embedding grid cells to capture relative spatial positions. Other approaches directly use coordinates, leveraging sequential models (i.e. RNNs) to process the encodings (Feng et al., 2018; Xue et al., 2021; Rao et al., 2020; Xu et al., 2018), but require strict sequential ordering and may overlook geometric relationships.

Polygon encoding has gained significant attention. Veer et al. (2018) employ elliptic Fourier descriptors to approximate polygon outlines and utilize bidirectional LSTM and 1D CNNs to encode vertex sequences. Mai et al. (2023) used a 1D ResNet architecture with circular padding for loop origin invariance. Other approaches use the non-uniform Fourier transform (NUFT) to map polygons to the spectral domain, converting them back into images via inverse Fourier transforms (IDFT), though this suffers from the limitations of grid-based approaches (Jiang et al., 2019a;b). Mai et al. (2023) refine this approach by omitting the IDFT. POLYGONGNN (Yu et al., 2024) encodes multipolygons, modeling their shape details and inter-polygonal relationships through heterogeneous visibility graphs.

While effective for specific geometry types, existing approaches are devised for specific geospatial objects. Encoding heterogeneous coordinate-based data remains a challenge, as current methods, in such cases, either use separate encoders for different object types, thereby adding complexity, or convert geometries into known formats (i.e., image, text), leading to a loss of spatial precision. This limitation is particularly critical for GeoAI models that aim to incorporate coordinate-based geospatial data as an additional modality (Zhang et al., 2024; Mai et al., 2024). POLY2VEC addresses this gap by uniformly encoding points, polylines, and polygons within the same framework, offering a level

of versatility not demonstrated by prior methods.

# 6. Conclusion and Future Work

We proposed POLY2VEC, a unified encoding framework for geospatial objects that preserves essential spatial properties, including topology, directionality, and distance. By outperforming object-specific baselines and improving downstream tasks like population prediction and land use inference, POLY2VEC demonstrates its versatility and effectiveness in GeoAI pipelines. Future work will explore extending POLY2VEC to higher-dimensional geometries, including 3D shapes, and its integration into Geo-Foundation models as a unified representation for coordinate data modalities.

# Impact Statement

This paper presents work whose goal is to advance the field of Machine Learning. There are many potential societal consequences of our work, none of which we feel must be specifically highlighted here. Our improved representation of 2D geometry for deep models could lead to more accurate, versatile GeoAI applications, leading to better understanding the Earth and improvements for the environment, transportation efficiency, and access equity.

# Acknowledgments

This research has been funded in part by the NIH award R01LM014026 and NSF award DMS-2428039. Any opinions, findings, and conclusions, or recommendations expressed in this material are those of the authors and do not necessarily reflect the views of the sponsors such as the NIH and NSF. J. Li and H. Lu's work was supported by Independent Research Fund Denmark (No. 1032-00481B). Part of H. Lu's work was done when the author was employed at Roskilde University.

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

# A. Appendix

### A.1. Geospatial Objects Definitions

**Definition 2** (Point). A point is a zero-dimensional geometric entity in $\mathbb{R}^2$, defined by a single coordinate $(x, y)$, where $x, y \in \mathbb{R}$. A point represents a specific location in the plane but has no extent, size, nor dimension.

**Definition 3** (Line Segment). A line segment is a one-dimensional geometric object in $\mathbb{R}^2$, defined as a straight line segment between two distinct endpoints $p_1 = (x_1, y_1)$ and $p_2 = (x_2, y_2)$.

**Definition 4** (Polyline). A polyline is a one-dimensional object in $\mathbb{R}^2$, represented by an array $P \in \mathbb{R}^{N \times 2}$, where each row is a point $p_i = (x_i, y_i)$. It consists of connected line segments formed by consecutive points $p_i$ and $p_{i+1}$ for $1 \le i < N$, with $p_1 \neq p_N$.

**Definition 5** (Polygon). A polygon is a two-dimensional geometric object in $\mathbb{R}^2$, represented as a closed sequence of points forming its boundary. It is defined by an array $P \in \mathbb{R}^{N \times 2}$, where each row corresponds to a point $(x_i, y_i) \in \mathbb{R}^2$ and $(x_1, y_1) = (x_N, y_N)$.

### A.2. Analytical Calculations of Fourier Transform

#### A.2.1. FOURIER TRANSFORM OF A POINT

By representing a point $p = (x_p, y_p) \in \mathbb{R}^2$ as a 2D Dirac delta function $f_p(x, y) = \delta(x - x_p, y - y_p)$ the Fourier transform of $f_p(x, y)$ can be derived as follows:

$$
\begin{aligned}
F_p(u, v) &= \mathscr{F}\{f_p(x, y)\} \\
&= \int_{-\infty}^{\infty} \int_{-\infty}^{\infty} f_p(x, y) e^{-j2\pi(ux+vy)} dx\, dy \\
&= \int_{-\infty}^{\infty} \int_{-\infty}^{\infty} \delta(x - x_p, y - y_p) e^{-j2\pi(ux+vy)} dx\, dy \\
&= e^{-j2\pi(x_p u + y_p v)}
\end{aligned}
$$

where $(u, v)$ are the frequency components.

#### A.2.2. FOURIER TRANSFORM OF A POLYLINE

**Canonical line segment.** We express the canonical line segment $l_c$ extending from $\mathbf{a} = (-\frac{1}{2}, 0)$ to $\mathbf{b} = (\frac{1}{2}, 0)$, as $f_{l_c}(x, y) = \text{rect}(x)\delta(y)$. where $\text{rect}(x)$ restricts the ridge to $|x| \le \frac{1}{2}$, and $\delta(y)$ represents a Dirac delta function along the $x$-axis. The Fourier transform of $f_{l_c}(x, y)$ is :

$$
\begin{aligned}
F_{l_c}(u, v) &= \mathscr{F}\{f_{l_c}(x, y)\} \\
&= \int_{-\infty}^{\infty} \int_{-\infty}^{\infty} f_{l_c}(x, y) e^{-j2\pi(ux+vy)} dx\, dy \\
&= \int_{-\infty}^{\infty} \int_{-\infty}^{\infty} \text{rect}(x)\delta(y) e^{-j2\pi(ux+vy)} dx\, dy
\end{aligned}
$$

Using the sifting property of the Dirac delta function, the integral over $y$ evaluates to the value of the integrand at $y = 0$:

$$
\begin{aligned}
F_{l_c}(u, v) &= \int_{-\infty}^{\infty} \text{rect}(x) e^{-j2\pi ux} e^{-j2\pi v(0)} dx \\
&= \int_{-\infty}^{\infty} \text{rect}(x) e^{-j2\pi ux} dx \\
&= \text{sinc}(u)
\end{aligned}
$$

where $(u, v)$ are the frequency components and $v = 0$.

**Arbitrary line segment.** We consider an arbitrary line segment $l$ with endpoints $\mathbf{q} = (x_q, y_q)$ and $\mathbf{r} = (x_r, y_r)$, to compute the $F_l(u, v)$, we map it to the canonical line segment $l_c$ using affine transformation. For this purpose, we introduce an

auxiliary point $\mathbf{c} = (\frac{1}{2}, 1)$ at the structure of $l_c$ so that it is not colinear with $\mathbf{ab}$. This point maps to another auxiliary point $\mathbf{s}$ introduced at the arbitrary line segment $l$. The auxiliary point $\mathbf{s}$ is defined as $\mathbf{s} = \mathbf{r} + \mathbf{n}$, where $\mathbf{n} = (y_q - y_r, x_r - x_q)^{\top}$, representing a $90°$ clockwise rotation of the vector $\mathbf{r} - \mathbf{q}$. Note that the vectors $\mathbf{qr}$ and $\mathbf{rs}$ are the same length.

Given all the above, the affine transformation matrix $\mathbf{A}$ is defined as:

$$\mathbf{A} = \begin{bmatrix} a_1 & b_1 & c_1 \\ a_2 & b_2 & c_2 \\ 0 & 0 & 1 \end{bmatrix}$$

Then the values of $\mathbf{A}$ are computed as follows:

$$\mathbf{A}\,[\mathbf{q}\,\mathbf{r}\,\mathbf{s}] = [\mathbf{a}\,\mathbf{b}\,\mathbf{c}]$$
$$\mathbf{A} = [\mathbf{a}\,\mathbf{b}\,\mathbf{c}][\mathbf{q}\,\mathbf{r}\,\mathbf{s}]^{-1}$$
$$= \begin{bmatrix} -\frac{1}{2} & \frac{1}{2} & \frac{1}{2} \\ 0 & 0 & 1 \\ 0 & 0 & 1 \end{bmatrix} \begin{bmatrix} x_q & x_r & x_r + y_q - y_r \\ y_q & y_r & y_r + x_r - x_q \\ 1 & 1 & 1 \end{bmatrix}^{-1}$$
$$= D \begin{bmatrix} -x_q + x_r & -y_q + y_r & \frac{(x_q^2 + y_q^2 - x_r^2 - y_r^2)}{2} \\ y_q - y_r & -x_q + x_r & -y_q x_r + x_q y_r \\ 0 & 0 & \frac{1}{D} \end{bmatrix}$$

where

$$|D| = \det(\mathbf{A}) = \frac{1}{(x_q - x_r)^2 + (y_q - y_r)^2}$$

is the determinant of $\mathbf{A}$.

Following the affine Fourier transform property from Eq. (3), the Fourier transform of an arbitrary line segment $l$ with endpoints $(x_q, y_q)$ and $(x_r, y_r)$ is:

$$F_l(u, v) = \mathscr{F}\{f_{l_c}(x, y)\}$$
$$= \frac{1}{|\det(\mathbf{A})|} e^{-j2\pi \mathbf{c}^{\top} \mathbf{A}^{-\top} \mathbf{u}} F(\mathbf{A}^{-\top} \mathbf{u})$$

which can be rewritten as:

$$F_l(u, v) =$$
$$= \frac{1}{|D|} e^{-j2\pi(x_0 u + y_0 v)} F\left( \frac{b_2 u - a_2 v}{|D|}, -\frac{b_1 u + a_1 v}{|D|} \right) \tag{15}$$

where $x_0 = \frac{1}{|D|}(b_1 c_2 - b_2 c_1)$ and $y_0 = \frac{1}{|D|}(a_2 c_1 - a_1 c_2)$.

By substituting the specific values into Eq. (15), $F_l(u, v)$ can be simplified to:

$$F_l(u, v) = \frac{1}{(x_q - x_r)^2 + (y_q - y_r)^2} \left[ e^{-j2\pi\left(\frac{x_q + x_r}{2} u + \frac{y_q + y_r}{2} v\right)} \operatorname{sinc}\left((x_r - x_q)u + (y_r - y_q)v\right) \right]$$

### A.2.3. FOURIER TRANSFORM OF A POLYGON

**Isosceles canonical right triangle.** The canonical isosceles right triangle $\triangle_c$ with vertices $\mathbf{a} = (0, 0)$, $\mathbf{b} = (1, 0)$, and $\mathbf{c} = (1, 1)$, is represented by the function $f_{\triangle_c}(x, y)$ which equals 1 inside the triangle and 0 otherwise.

The Fourier transform of $f_{\Delta_c}(x, y)$ is computed as:

$$
\begin{aligned}
F_{\Delta_c}(u, v) &= \mathscr{F}\{f_{\Delta_c}(x, y)\} \\
&= \int \int f_{\Delta_c}(x, y) e^{-j2\pi(ux+vy)} \, dy \, dx \\
&= \int_0^1 \int_0^x e^{-j2\pi(ux+vy)} \, dy \, dx \\
&= \int_0^1 \frac{1}{-j2\pi v} \left( e^{-j2\pi(u+v)x} - e^{-j2\pi ux} \right) dx \\
&= \frac{1}{-j2\pi v} \left[ \int_0^1 e^{-j2\pi(u+v)x} \, dx - \int_0^1 e^{-j2\pi ux} \, dx \right] \\
&= \frac{1}{4\pi^2 v(u+v)} \left[ (u+v) e^{-j2\pi u} - u e^{-j2\pi(u+v)} - v \right]
\end{aligned}
\tag{16}
$$

Using Euler's formula ($e^{j\theta} = \cos\theta + j\sin\theta$), we can expand Eq. (16) to:

$$
F_{\Delta_c}(u, v) = \frac{1}{4\pi^2 uv(u+v)} \left[ \left( (u+v)\cos(2\pi u) - u\cos(2\pi(u+v)) - v \right) - j\left( (u+v)\sin(2\pi u) - u\sin(2\pi(u+v)) \right) \right]
$$

This equation is undefined for some values of $(u, v)$. We present the Fourier transform for each special case:

- $F_{\Delta_c}(0, 0) = \dfrac{1}{2}$

- $F_{\Delta_c}(0, v) = -\dfrac{1}{4\pi^2 v^2} \left( j2\pi v + \cos(2\pi v) - j\sin(2\pi v) - 1 \right)$

- $F_{\Delta_c}(u, 0) = \dfrac{1}{4\pi^2 u^2} \left[ \left( \cos(2\pi u) + 2\pi u \sin(2\pi u) - 1 \right) - j\left( \sin(2\pi u) - 2\pi u \cos(2\pi u) \right) \right]$

- $F_{\Delta_c}(-v, v) = -\dfrac{1}{4\pi^2 v^2} \left( -j2\pi v + \cos(2\pi v) + j\sin(2\pi v) - 1 \right)$

**Arbitrary triangle.** We calculate the Fourier transform of an arbitrary triangle $\Delta$, with vertices $\mathbf{q}, \mathbf{r}, \mathbf{s}$ by using the affine transformation property. To that extent the affine transformation matrix $\mathbf{A}$ is defined as:

$$
\mathbf{A} = \begin{bmatrix} a_1 & b_1 & c_1 \\ a_2 & b_2 & c_2 \\ 0 & 0 & 1 \end{bmatrix}
$$

Then the values of $\mathbf{A}$ are computed as follows:

$$
\mathbf{A}\,[\mathbf{q}\,\mathbf{r}\,\mathbf{s}] = [\mathbf{a}\,\mathbf{b}\,\mathbf{c}]
$$

$$
\mathbf{A} = [\mathbf{a}\,\mathbf{b}\,\mathbf{c}][\mathbf{q}\,\mathbf{r}\,\mathbf{s}]^{-1}
$$

$$
= \begin{bmatrix} 0 & 1 & 1 \\ 0 & 0 & 1 \\ 1 & 1 & 1 \end{bmatrix} \begin{bmatrix} x_q & x_r & x_r + y_q - y_r \\ y_q & y_r & y_r + x_r - x_q \\ 1 & 1 & 1 \end{bmatrix}^{-1} = D \begin{bmatrix} y_s - y_r & x_r - x_s & y_q(x_s - x_r) + x_q(y_r - y_s) \\ y_q - y_r & x_r - x_q & x_q y_r - y_q x_r \\ 0 & 0 & D \end{bmatrix}
$$

where

$$
|D| = \frac{1}{x_q(y_r - y_s) + x_r(y_s - y_q) + x_s(y_q - y_r)}
$$

is the determinant of $\mathbf{A}$.

If the area of $\Delta$ is $\alpha$, then $D = \frac{1}{2\alpha}$.

Finally the Fourier transform $F_\Delta(u, v)$ can be calculated by substituting the affine transform parameters into Eq. (3).

For the case of $(0, 0)$ we get that : $F_\Delta(0, 0) = \frac{1}{D} F_{\Delta_c}(0, 0) = \frac{1}{2D} = \alpha$, which is the area of $\Delta$.

### A.3. Frequency Sampling Strategy

#### A.3.1. DETAILS ON THE GEOMETRIC FREQUENCY SAMPLING

We sample frequencies as a geometric series to balance the contribution of low and high-frequency frequency components. Formally,

$$f_i = f_{\min} \cdot \rho_i, \quad i = 0, 1, \ldots, W - 1$$

where $f_i$ is the $i$-th frequency, $f_{\min}$, $f_{\max}$ correspond to the minimum and maximum frequencies and $W$ is the number of sampled frequencies in each dimension. $\rho_i$ is the step ratio and is defined as $\rho_i = \left(\frac{f_{max}}{f_{min}}\right)^{\frac{1}{(W-1)}}$.

Using this sequence, we construct a 2D meshgrid of frequencies, denoted as (U,V), centered around zero. Due to the Hermitian symmetry property of the Fourier transform, we only compute frequencies for half of the plane.

While uniform sampling is an alternative, previous studies suggest geometric sampling is better suited for tasks like ours, as it naturally balances the significance of low- and high-frequency components (Mai et al., 2023).

#### A.3.2. ADDITIONAL STRATEGIES

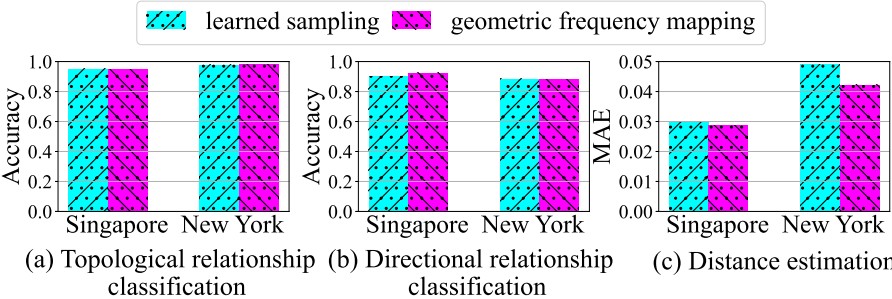

Figure 5: The effect of frequency sampling strategy on point-polygon pairs.

To investigate whether learning the frequency values would improve performance, we conducted an experiment where the frequencies were treated as learnable parameters and optimized alongside the model. Our results are reported in Figure 5. We observe that learning the frequencies does not yield significant improvements over fixing the frequencies in any of the tasks. This suggests that the geometric sampling approach is sufficiently effective for balancing low- and high-frequency contributions, and learning the frequencies does not provide additional benefits for the tasks considered.

### A.4. Dataset Details

We utilized publicly available OpenStreetMap (OSM) datasets for Singapore and New York, obtained from Geofabrik[9] in `.osm.pbf` format. Geospatial objects, including POIs, roads, and buildings, were extracted using OSM-specific tags (amenity, shop, tourism, leisure for POIs, motorway, trunk, primary, secondary for roads, and building for buildings). Region partitions were derived from Singapore Subzones[10] and NYC Census Tracts[11]. Dataset statistics are presented in Table 5.

| City | # POIs | # roads | # buildings | # regions |
|------|--------|---------|-------------|-----------|
| Singapore | 4,347 | 45,634 | 109,877 | 304 |
| New York | 14,943 | 139,512 | 1,153,088 | 2,324 |

Table 5: Statistics of the Singapore and New York datasets.

---

[9]https://download.geofabrik.de/
[10]https://data.gov.sg/collections/1749/view
[11]https://www.nyc.gov/site/planning/data-maps/open-data/census-download-metadata.page

Labels for the **land use classification** task were sourced from the Singapore Master Plan 2019[12] and NYC MapPLUTO[13]. Following previous approaches (Li et al., 2023), we merge the fine-grained land use classes into five major categories, including Residential, Industrial, Commercial, Open Space, and Others. **Population estimation** labels were obtained from WorldPop[14] for both cities.

For the remaining tasks, the labels are generated manually. Specifically, for the **topological classification** task, the number of relationships depends on the types of objects being compared. Point/polyline, point/polygon, and polyline/polyline pairs can belong to one of two classes: *disjoint* or *not disjoint*. Polyline/polygon pairs, however, have four distinct relationship classes, while polygon/polygon pairs include six classes, following the DE9IM model. To eliminate redundancy, we remove equivalent relationships such as *within* and *contains*, keeping only one representative relationship from each pair of equivalents. To create a balanced dataset across all relationship classes, we generate geometry pairs by slightly adjusting the positions of the original geospatial objects and randomly selecting 5,000 pairs for each class within a group.

For the **directional relationship classification** task, we classify the spatial relationships between two geometries into one of 16 compass directions based on their angular relationship. These 16 classes are derived from the cardinal and intercardinal directions: *north*, *northeast*, *east*, *southeast*, *south*, *southwest*, *west*, *northwest*, and their boundary counterparts (e.g., *north-northeast*, *east-northeast*). Labels are computed based on the relative orientation of the geometries' centroids. Similar to the topological classification task, we randomly select 5,000 pairs for each directional class to ensure a balanced dataset.

For the **distance estimation** task, labels are computed using the actual spatial distance between the centroids of the two geometries. The spatial distance is calculated using Euclidean distance for planar geometries, for topological and directional relationship classification. We randomly select 10,000 geometry pairs for this task.

## A.5. Baselines

We now describe the baseline methods used to evaluate POLY2VEC.

### 1. Point encoders

• DIRECT: Feeds directly the geometry's input coordinates to the downstream model, without any encoding mechanism.

• TILE: Partitions the study area into a uniform grid with cells of size $c$. Each grid cell is assigned an embedding, which serves as the encoding for the points assigned to that cell (Berg et al., 2014; Adams et al., 2015; Tang et al., 2015).

• WRAP: Uses a wrapping mechanism $[\sin(\pi p); \cos(\pi p)]$ to encode a point $p$ (Mac Aodha et al., 2019).

• GRID: Follows the Transformer's position encoding model (Vaswani, 2017), representing spatial positions through multi-scale sine and cosine transformations. At each scale $s$, the encoding is given by $PE_s^{(g)}(p) = \left[\cos\left(\frac{p}{\lambda_{\min} \cdot g^{\frac{s}{S-1}}}\right), \sin\left(\frac{p}{\lambda_{\min} \cdot g^{\frac{s}{S-1}}}\right)\right]$, where $g = \frac{\lambda_{\max}}{\lambda_{\min}}$ controls the frequency range. The final encoding concatenates these multi-scale representations, capturing spatial structures across different resolutions (Mai et al., 2020).

• THEORY: Encodes spatial positions using dot products with unit vectors separated by $120°$. At each scale $s$, the encoding is given by $PE_{s,j}^{(t)}(p) = \left[\cos\left(\frac{\langle p, a_j \rangle}{\lambda_{\min} \cdot g^{\frac{s}{S-1}}}\right), \sin\left(\frac{\langle p, a_j \rangle}{\lambda_{\min} \cdot g^{\frac{s}{S-1}}}\right)\right] \quad \forall j \in \{1, 2, 3\}$, where $a_1 = [1, 0]^T$, $a_2 = [-\frac{1}{2}, \frac{\sqrt{3}}{2}]^T$, and $a_3 = [-\frac{1}{2}, -\frac{\sqrt{3}}{2}]^T$ are unit vectors spaced at $120°$. The final encoding concatenates these multi-scale representations across all vectors (Mai et al., 2020).

### 2. Polyline encoders

• T2VEC: First uniformly partitions the whole space into grid cells, and map each trajectory point into the grid cell. Through this tokenization, each trajectory is converted to a sequence of grid cell IDs. Then adopts a GRU encoder to encode the sequence and an end-to-end training paradigm that amis to reconstruct the original trajectories from the distorted/downsampled ones (Li et al., 2018a).

### 3. Polygon encoders

---

[12] https://data.gov.sg/dataset/master-plan-2019-land-use-laye
[13] https://www.nyc.gov/site/planning/data-maps/open-data/dwn-pluto-mappluto.page
[14] https://hub.worldpop.org/geodata/listing?id=77

- RESNET1D: Adapts the 1D variant of the Residual Network (ResNet) architecture, incorporating circular padding to effectively encode the exterior vertices of polygons (Mai et al., 2023).

- NUFTSPEC: Transforms polygons into the spectral domain using the Non-Uniform Fourier Transformation (NUFT) and $j$-simplex meshes and then learns polygon embeddings from these spectral features using MLPs (Mai et al., 2023).

## A.6. Hyperparameter Configuration

The coordinates of the input geometries are normalized to lie within the range $[-1, 1] \times [-1, 1]$, based on the bounding box of the corresponding area of interest. We set the minimum frequency $f_{\min} = 0.1$, the maximum frequency $f_{\max} = 1.0$ and $W = 10$, resulting in 210 frequencies. We set the final size of the geometry embedding $\mathbf{v}$ to $d = 32$. All the MLPs consist of two layers with ReLU activation functions.

### A.6.1. HYPERPARAMETERS OF SPATIAL REASONING TASKS.

For training on the spatial reasoning tasks, we utilize the AdamW optimizer and set the learning rate $lr = 10^{-4}$ and weight decay $wd = 10^{-8}$. The batch size is set to 128, and the downstream models were trained for 20 epochs. The training, validation, and testing ratios for the datasets corresponding to these tasks is 60:20:20. All experiments were run 5 times and we report average performances and standard deviation.

### A.6.2. HYPERPARAMETERS OF GEOAI TASKS.

We follow the same hyperparameters as presented by Li et al. (2023), to keep our comparison consistent.

### A.6.3. HYPERPARAMETERS OF OTHER BASELINES.

The implementation of baselines follows the corresponding papers, along which each method's specific hyperparameters. The rest of hyperparameters related to downstream tasks are kept consistent with our approach.

## A.7. Experimental Environment

Our experiments are performed on a cluster node equipped with an 18-core Intel i9-9980XE CPU, 125 GB of memory, and two 11 GB NVIDIA GeForce RTX 2080 Ti GPUs. Furthermore, all neural network models are implemented based on PyTorch version 2.3.0 with CUDA 11.8 using Python version 3.9.19.

## A.8. Training Details of Evaluation Tasks

We use cross entropy loss to train the downstream model on the topological and directional relationship classification tasks. The loss is defined as:

$$\mathcal{L}_{\text{CE}(\theta)} = -\frac{1}{N} \sum_{i=1}^{N} \sum_{c=1}^{C} y_{i,c} \log(\hat{y}_{i,c}),$$

where $N$ is the number of samples, $C$ is the number of classes ($C = 2$ for binary classification), $y_{i,c} \in \{0, 1\}$ is the one-hot encoded ground-truth label for class $c$, and $\hat{y}_{i,c} \in [0, 1]$ is the predicted probability for class $c$.

For the distance preservation task, the model is evaluated using the mean squared error (MSE) loss, defined as:

$$\mathcal{L}_{\text{MSE}(\theta)} = \frac{1}{N} \sum_{i=1}^{N} (y_i - \hat{y}_i)^2,$$

where $y_i$ is the ground-truth distance for the $i$-th sample, and $\hat{y}_i$ is the predicted distance.

We note that for the population prediction and land use classification tasks, POLY2VEC is used as input to the pretrained urban region representation model REGIONDCL (Li et al., 2023), and thus we follow the same training and evaluation procedure as was originally presented by the authors.

### A.9. Further Details on Poly2Vec Integration into an End-to-End GeoAI Pipeline

In this section, we describe in more detail how Poly2Vec is integrated into RegionDCL.

We utilize RegionDCL as an end-to-end GeoAI pipeline to demonstrate the utility of Poly2Vec. Poly2Vec is used as the input encoding in RegionDCL, replacing its original input representation. RegionDCL originally rasterizes OSM building footprints, converting coordinate data into image inputs so it can leverage convolutional encoders like ResNet-18. This rasterization leads to the loss of important spatial information, such as the absolute location of each building. To mitigate this, RegionDCL introduces a distance-biased transformer encoder, where the bias term consists of pairwise distances between buildings and POIs to reintroduce spatial context.

In our experiments, we (1) replaced the inputs with Poly2Vec encodings, and (2) replaced the distance-biased transformer encoder with a standard transformer encoder, because our new inputs from (1) capture the necessary spatial information. The fact that Poly2Vec improves performance even without the distance bias demonstrates its ability to inherently retain spatial and positional information.

### A.10. Supplementary Results on Topological Relationships Classification

Table 6: Overall model Performance on topological relationship classification. **Best** and second best are highlighted.

| Metric | Methods | Singapore | | | | | New York | | | | |
|---|---|---|---|---|---|---|---|---|---|---|---|
| | | point-polyline | point-polygon | polyline-polyline | polyline-polygon | polygon-polygon | point-polyline | point-polygon | polyline-polyline | polyline-polygon | polygon-polygon |
| Precision | ResNet1D | - | - | - | - | $0.398_{0.018}$ | - | - | - | - | $0.421_{0.051}$ |
| | NUFTspec | - | - | - | - | $0.588_{0.041}$ | - | - | - | - | $0.562_{0.032}$ |
| | T2vec | - | - | $0.768_{0.021}$ | - | - | - | - | $0.745_{0.012}$ | - | - |
| | Direct | $0.859_{0.007}$ | $0.831_{0.017}$ | $0.637_{0.032}$ | $0.415_{0.037}$ | $0.328_{0.04}$ | $0.835_{0.032}$ | $0.933_{0.007}$ | $0.661_{0.032}$ | $0.498_{0.003}$ | $0.439_{0.024}$ |
| | Tile | $0.735_{0.039}$ | $0.705_{0.056}$ | $0.505_{0.007}$ | $0.490_{0.006}$ | $0.439_{0.005}$ | $0.664_{0.018}$ | $0.789_{0.005}$ | $0.502_{0.009}$ | $0.494_{0.074}$ | $0.418_{0.005}$ |
| | Wrap | $0.874_{0.011}$ | $0.865_{0.015}$ | $0.645_{0.009}$ | $0.453_{0.028}$ | $0.405_{0.010}$ | $0.879_{0.015}$ | $0.915_{0.006}$ | $0.655_{0.013}$ | $0.586_{0.005}$ | $0.405_{0.010}$ |
| | Grid | $0.799_{0.037}$ | $0.841_{0.010}$ | $0.626_{0.027}$ | $0.405_{0.066}$ | $0.288_{0.013}$ | $0.768_{0.034}$ | $0.904_{0.015}$ | $0.658_{0.014}$ | $0.513_{0.012}$ | $0.355_{0.017}$ |
| | Theory | $0.903_{0.037}$ | $0.874_{0.004}$ | $0.651_{0.009}$ | $0.432_{0.018}$ | $0.478_{0.023}$ | $0.886_{0.044}$ | $0.893_{0.017}$ | $0.718_{0.007}$ | $0.602_{0.008}$ | $0.431_{0.009}$ |
| | Poly2Vec | **$0.913_{0.007}$** | **$0.924_{0.017}$** | **$0.779_{0.001}$** | **$0.506_{0.013}$** | **$0.694_{0.007}$** | **$0.921_{0.016}$** | **$0.979_{0.021}$** | **$0.745_{0.002}$** | **$0.631_{0.017}$** | **$0.698_{0.006}$** |
| Recall | ResNet1D | - | - | - | - | $0.455_{0.011}$ | - | - | - | - | $0.452_{0.035}$ |
| | NUFTspec | - | - | - | - | $0.572_{0.032}$ | - | - | - | - | $0.592_{0.029}$ |
| | T2vec | - | - | $0.732_{0.024}$ | - | - | - | - | $0.718_{0.032}$ | - | - |
| | Direct | $0.792_{0.012}$ | $0.838_{0.027}$ | $0.997_{0.019}$ | $0.414_{0.031}$ | $0.450_{0.014}$ | $0.838_{0.035}$ | $0.888_{0.004}$ | $0.987_{0.22}$ | $0.497_{0.003}$ | $0.431_{0.003}$ |
| | Tile | $0.894_{0.035}$ | $0.695_{0.074}$ | $1.0_{0.001}$ | $0.463_{0.008}$ | $0.413_{0.004}$ | $0.659_{0.009}$ | $0.769_{0.011}$ | $1.00_{0.001}$ | $0.499_{0.039}$ | $0.405_{0.004}$ |
| | Wrap | $0.903_{0.005}$ | $0.901_{0.033}$ | $0.992_{0.001}$ | $0.477_{0.012}$ | $0.380_{0.006}$ | $0.894_{0.030}$ | $0.842_{0.031}$ | $0.986_{0.005}$ | $0.551_{0.008}$ | $0.380_{0.006}$ |
| | Grid | $0.921_{0.035}$ | $0.848_{0.014}$ | $0.980_{0.016}$ | $0.465_{0.007}$ | $0.339_{0.013}$ | $0.933_{0.045}$ | $0.881_{0.004}$ | $0.995_{0.002}$ | $0.514_{0.012}$ | $0.382_{0.035}$ |
| | Theory | $0.986_{0.028}$ | $0.933_{0.007}$ | $0.972_{0.012}$ | $0.451_{0.012}$ | $0.467_{0.015}$ | $0.923_{0.044}$ | $0.912_{0.017}$ | $0.782_{0.007}$ | $0.615_{0.008}$ | $0.412_{0.009}$ |
| | Poly2Vec | **$1.0_{0.000}$** | **$0.974_{0.023}$** | **$1.0_{0.000}$** | **$0.498_{0.007}$** | **$0.697_{0.003}$** | **$1.0_{0.000}$** | **$0.989_{0.032}$** | **$1.0_{0.000}$** | **$0.638_{0.009}$** | **$0.697_{0.007}$** |
| F1 | ResNet1D | - | - | - | - | $0.399_{0.017}$ | - | - | - | - | $0.399_{0.041}$ |
| | NUFTspec | - | - | - | - | $0.574_{0.013}$ | - | - | - | - | $0.581_{0.021}$ |
| | T2vec | - | - | $0.732_{0.002}$ | - | - | - | - | $0.741_{0.007}$ | - | - |
| | Direct | $0.824_{0.006}$ | $0.834_{0.031}$ | $0.777_{0.022}$ | $0.402_{0.027}$ | $0.314_{0.014}$ | $0.836_{0.004}$ | $0.910_{0.003}$ | $0.792_{0.027}$ | $0.463_{0.003}$ | $0.403_{0.013}$ |
| | Tile | $0.805_{0.013}$ | $0.694_{0.017}$ | $0.671_{0.004}$ | $0.412_{0.009}$ | $0.384_{0.005}$ | $0.661_{0.008}$ | $0.779_{0.004}$ | $0.668_{0.008}$ | $0.453_{0.061}$ | $0.369_{0.003}$ |
| | Wrap | $0.888_{0.005}$ | $0.882_{0.009}$ | $0.781_{0.008}$ | $0.450_{0.020}$ | $0.339_{0.006}$ | $0.886_{0.009}$ | $0.876_{0.019}$ | $0.787_{0.010}$ | $0.517_{0.005}$ | $0.339_{0.006}$ |
| | Grid | $0.855_{0.007}$ | $0.844_{0.002}$ | $0.764_{0.015}$ | $0.411_{0.026}$ | $0.267_{0.018}$ | $0.842_{0.032}$ | $0.892_{0.006}$ | $0.792_{0.009}$ | $0.463_{0.046}$ | $0.322_{0.038}$ |
| | Theory | $0.938_{0.014}$ | $0.903_{0.004}$ | $0.788_{0.007}$ | $0.438_{0.012}$ | $0.425_{0.006}$ | $0.883_{0.044}$ | $0.891_{0.017}$ | $0.726_{0.007}$ | $0.549_{0.059}$ | $0.419_{0.009}$ |
| | Poly2Vec | **$0.955_{0.011}$** | **$0.948_{0.008}$** | **$0.831_{0.002}$** | **$0.483_{0.013}$** | **$0.682_{0.003}$** | **$0.959_{0.008}$** | **$0.984_{0.012}$** | **$0.854_{0.002}$** | **$0.588_{0.012}$** | **$0.679_{0.005}$** |

### A.11. Supplementary Results on Directional Relationship Classification

Table 7: Overall model Performance on directional relationship classification. **Best** and second best are highlighted.

| Metric | Methods | Singapore | | | | | | New York | | | | | |
|---|---|---|---|---|---|---|---|---|---|---|---|---|---|
| | | point-point | point-polyline | point-polygon | polyline-polyline | polyline-polygon | polygon-polygon | point-point | point-polyline | point-polygon | polyline-polyline | polyline-polygon | polygon-polygon |
| Precision | NUFTRESNET | - | - | - | - | - | $0.828_{0.009}$ | - | - | - | - | - | $0.783_{0.010}$ |
| | NUFTSPEC | - | - | - | - | - | **$0.832_{0.021}$** | - | - | - | - | - | $0.715_{0.014}$ |
| | T2VEC | - | - | - | $0.227_{0.021}$ | - | - | - | - | - | $0.232_{0.012}$ | - | - |
| | DIRECT | $0.882_{0.006}$ | $0.846_{0.006}$ | $0.847_{0.005}$ | $0.825_{0.002}$ | $0.813_{0.005}$ | $0.765_{0.014}$ | $0.880_{0.003}$ | $0.767_{0.004}$ | $0.843_{0.002}$ | $0.687_{0.003}$ | $0.794_{0.003}$ | $0.774_{0.001}$ |
| | TILE | $0.259_{0.001}$ | $0.260_{0.026}$ | $0.286_{0.038}$ | $0.370_{0.005}$ | $0.466_{0.001}$ | $0.415_{0.010}$ | $0.293_{0.001}$ | $0.279_{0.013}$ | $0.322_{0.005}$ | $0.280_{0.005}$ | $0.496_{0.002}$ | $0.376_{0.026}$ |
| | WRAP | $0.863_{0.003}$ | $0.810_{0.007}$ | $0.806_{0.004}$ | $0.790_{0.002}$ | $0.835_{0.002}$ | $0.789_{0.001}$ | $0.809_{0.004}$ | $0.684_{0.002}$ | $0.759_{0.016}$ | $0.610_{0.021}$ | $0.781_{0.001}$ | $0.667_{0.007}$ |
| | GRID | $0.884_{0.007}$ | $0.733_{0.007}$ | $0.775_{0.002}$ | $0.708_{0.001}$ | $0.653_{0.015}$ | $0.545_{0.144}$ | $0.872_{0.002}$ | $0.605_{0.001}$ | $0.670_{0.040}$ | $0.441_{0.003}$ | $0.766_{0.003}$ | $0.514_{0.074}$ |
| | THEORY | $0.908_{0.017}$ | $0.872_{0.012}$ | $0.863_{0.004}$ | $0.815_{0.012}$ | $0.838_{0.006}$ | $0.729_{0.044}$ | $0.881_{0.017}$ | $0.774_{0.007}$ | $0.809_{0.008}$ | $0.692_{0.009}$ | $0.789_{0.005}$ | $0.538_{0.012}$ |
| | POLY2VEC | **$0.928_{0.016}$** | **$0.942_{0.012}$** | **$0.918_{0.004}$** | **$0.911_{0.013}$** | **$0.898_{0.021}$** | $0.830_{0.007}$ | **$0.921_{0.006}$** | **$0.889_{0.016}$** | **$0.875_{0.004}$** | **$0.889_{0.013}$** | **$0.853_{0.007}$** | **$0.792_{0.009}$** |
| Recall | NUFTRESNET | - | - | - | - | - | $0.819_{0.010}$ | - | - | - | - | - | $0.747_{0.010}$ |
| | NUFTSPEC | - | - | - | - | - | $0.792_{0.003}$ | - | - | - | - | - | $0.685_{0.004}$ |
| | T2VEC | - | - | - | $0.216_{0.023}$ | - | - | - | - | - | $0.253_{0.032}$ | - | - |
| | DIRECT | $0.879_{0.006}$ | $0.841_{0.006}$ | $0.845_{0.006}$ | $0.820_{0.002}$ | $0.830_{0.005}$ | $0.752_{0.017}$ | $0.877_{0.004}$ | $0.766_{0.005}$ | $0.836_{0.002}$ | $0.653_{0.007}$ | $0.784_{0.004}$ | $0.694_{0.003}$ |
| | TILE | $0.253_{0.001}$ | $0.269_{0.002}$ | $0.273_{0.008}$ | $0.324_{0.001}$ | $0.454_{0.001}$ | $0.395_{0.003}$ | $0.248_{0.001}$ | $0.257_{0.004}$ | $0.316_{0.005}$ | $0.217_{0.001}$ | $0.466_{0.001}$ | $0.348_{0.012}$ |
| | WRAP | $0.861_{0.003}$ | $0.804_{0.009}$ | $0.803_{0.004}$ | $0.782_{0.003}$ | $0.831_{0.002}$ | $0.779_{0.001}$ | $0.810_{0.004}$ | $0.669_{0.001}$ | $0.759_{0.016}$ | $0.598_{0.018}$ | $0.772_{0.002}$ | $0.602_{0.006}$ |
| | GRID | $0.882_{0.002}$ | $0.729_{0.007}$ | $0.772_{0.002}$ | $0.699_{0.001}$ | $0.641_{0.016}$ | $0.533_{0.139}$ | $0.868_{0.002}$ | $0.590_{0.002}$ | $0.647_{0.050}$ | $0.437_{0.002}$ | $0.752_{0.003}$ | $0.483_{0.078}$ |
| | THEORY | $0.883_{0.024}$ | $0.867_{0.009}$ | $0.855_{0.004}$ | $0.863_{0.012}$ | $0.502_{0.012}$ | $0.897_{0.014}$ | $0.783_{0.021}$ | $0.791_{0.007}$ | $0.823_{0.008}$ | $0.709_{0.009}$ | $0.803_{0.005}$ | $0.567_{0.012}$ |
| | POLY2VEC | **$0.946_{0.017}$** | **$0.947_{0.021}$** | **$0.933_{0.011}$** | **$0.903_{0.008}$** | **$0.838_{0.022}$** | $0.826_{0.007}$ | **$0.923_{0.017}$** | **$0.894_{0.012}$** | **$0.886_{0.024}$** | **$0.878_{0.013}$** | **$0.875_{0.011}$** | **$0.793_{0.012}$** |
| F1 | NUFTRESNET | - | - | - | - | - | $0.821_{0.010}$ | - | - | - | - | - | $0.756_{0.010}$ |
| | NUFTSPEC | - | - | - | - | - | $0.802_{0.028}$ | - | - | - | - | - | $0.667_{0.023}$ |
| | T2VEC | - | - | - | $0.219_{0.007}$ | - | - | - | - | - | $0.252_{0.018}$ | - | - |
| | DIRECT | $0.880_{0.006}$ | $0.841_{0.006}$ | $0.845_{0.006}$ | $0.821_{0.002}$ | $0.840_{0.005}$ | $0.754_{0.016}$ | $0.876_{0.004}$ | $0.769_{0.005}$ | $0.838_{0.002}$ | $0.656_{0.009}$ | $0.784_{0.004}$ | $0.712_{0.002}$ |
| | TILE | $0.215_{0.001}$ | $0.226_{0.005}$ | $0.247_{0.015}$ | $0.309_{0.003}$ | $0.447_{0.001}$ | $0.388_{0.004}$ | $0.236_{0.001}$ | $0.212_{0.011}$ | $0.288_{0.012}$ | $0.193_{0.002}$ | $0.439_{0.002}$ | $0.339_{0.018}$ |
| | WRAP | $0.861_{0.003}$ | $0.804_{0.009}$ | $0.803_{0.004}$ | $0.782_{0.002}$ | $0.831_{0.002}$ | $0.780_{0.001}$ | $0.809_{0.004}$ | $0.668_{0.002}$ | $0.752_{0.002}$ | $0.590_{0.021}$ | $0.769_{0.002}$ | $0.613_{0.005}$ |
| | GRID | $0.882_{0.007}$ | $0.728_{0.007}$ | $0.772_{0.002}$ | $0.698_{0.001}$ | $0.640_{0.017}$ | $0.530_{0.150}$ | $0.868_{0.002}$ | $0.588_{0.002}$ | $0.649_{0.049}$ | $0.409_{0.003}$ | $0.749_{0.003}$ | $0.460_{0.077}$ |
| | THEORY | $0.903_{0.015}$ | $0.852_{0.009}$ | $0.855_{0.004}$ | $0.842_{0.012}$ | $0.845_{0.006}$ | $0.741_{0.044}$ | $0.884_{0.017}$ | $0.752_{0.007}$ | $0.812_{0.008}$ | $0.668_{0.009}$ | $0.756_{0.025}$ | $0.537_{0.22}$ |
| | POLY2VEC | **$0.928_{0.015}$** | **$0.927_{0.032}$** | **$0.918_{0.029}$** | **$0.901_{0.017}$** | **$0.899_{0.016}$** | **$0.827_{0.022}$** | **$0.892_{0.012}$** | **$0.883_{0.014}$** | **$0.903_{0.013}$** | **$0.877_{0.004}$** | **$0.832_{0.003}$** | **$0.769_{0.019}$** |

### A.12. Supplementary Results on Distance Estimation

Table 8: Overall model performance on distance estimation. **Best** and second best are highlighted.

| Dataset | Model | point-point | point-polyline | point-polygon |
|---|---|---|---|---|
| Singapore | DIRECT | $0.088_{\pm0.041}$ | $0.093_{\pm0.013}$ | $0.084_{\pm0.021}$ |
| | TILE | $0.252_{\pm0.002}$ | $0.177_{\pm0.007}$ | $0.157_{\pm0.001}$ |
| | WRAP | $0.085_{\pm0.009}$ | $0.106_{\pm0.012}$ | $0.102_{\pm0.007}$ |
| | GRID | $0.087_{\pm0.006}$ | $0.107_{\pm0.003}$ | $0.108_{\pm0.002}$ |
| | THEORY | $0.065_{\pm0.019}$ | $0.083_{\pm0.027}$ | $0.079_{\pm0.028}$ |
| | POLY2VEC | **$0.016_{\pm0.001}$** | **$0.043_{\pm0.011}$** | **$0.029_{\pm0.009}$** |
| New York | DIRECT | $0.075_{\pm0.017}$ | $0.126_{\pm0.041}$ | $0.115_{\pm0.033}$ |
| | TILE | $0.271_{\pm0.005}$ | $0.170_{\pm0.004}$ | $0.189_{\pm0.004}$ |
| | WRAP | $0.106_{\pm0.003}$ | $0.148_{\pm0.001}$ | $0.146_{\pm0.009}$ |
| | GRID | $0.073_{\pm0.001}$ | $0.124_{\pm0.004}$ | $0.118_{\pm0.011}$ |
| | THEORY | $0.068_{\pm0.008}$ | $0.089_{\pm0.074}$ | $0.102_{\pm0.061}$ |
| | POLY2VEC | **$0.030_{\pm0.007}$** | **$0.049_{\pm0.004}$** | **$0.042_{\pm0.021}$** |

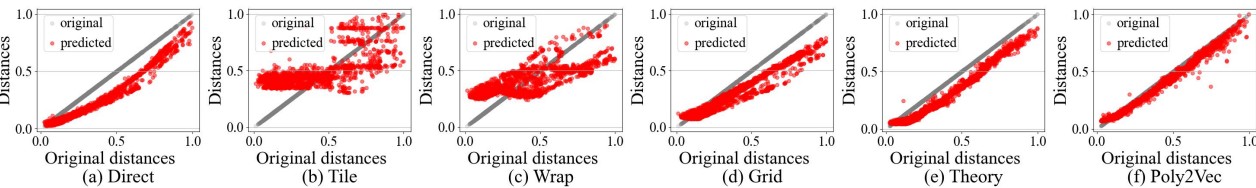

Figure 6: Distance scatters of point-polygon pairs on NewYork dataset for different encoders.

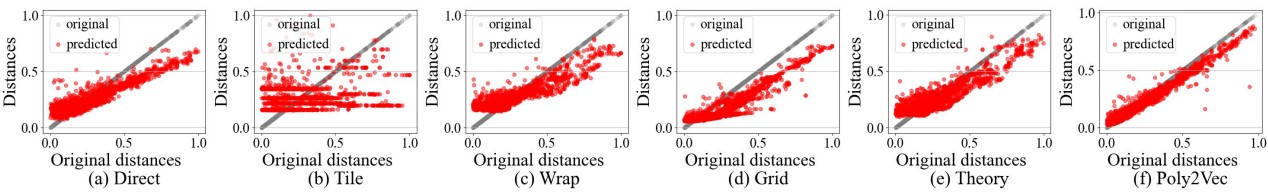

Figure 7: Distance scatters of point-polyline pairs on Singapore dataset for different encoders.

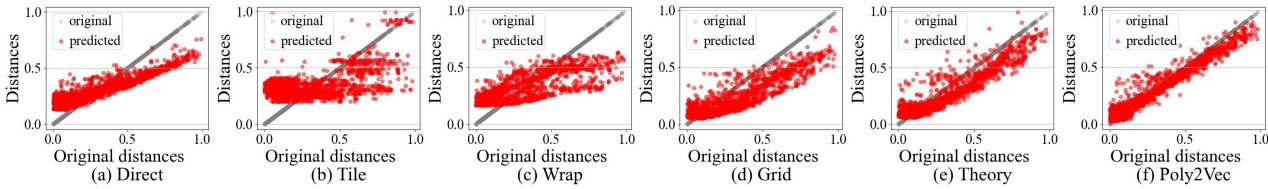

Figure 8: Distance scatters of point-polyline pairs on NewYork dataset for different encoders.

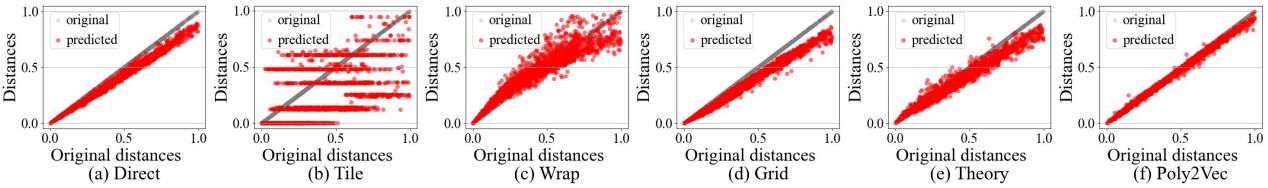

Figure 9: Distance scatters of point-point pairs on Singapore dataset for different encoders.

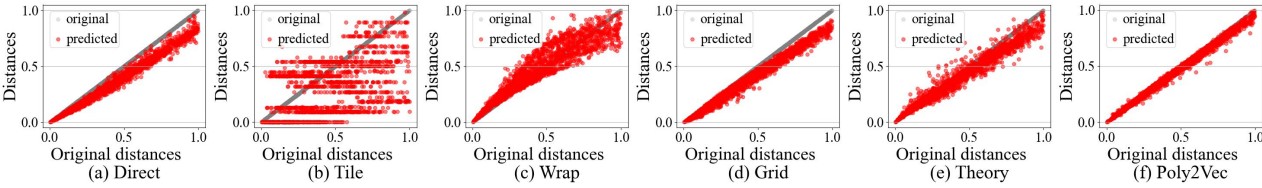

Figure 10: Distance scatters of point-point pairs on NewYork dataset for different encoders.

## A.13. Supplementary Ablation Experiments

### A.13.1. EFFECT OF LEARNED FUSION MODULE

We've shown the effect of learned fusion on point-polygon tasks in Section 4.3. We demonstrate its effect on the rest of spatial reasoning tasks in Figures 12a, 12b, 11a, and 11b. We again observe similar trends as reported in the main evaluation.

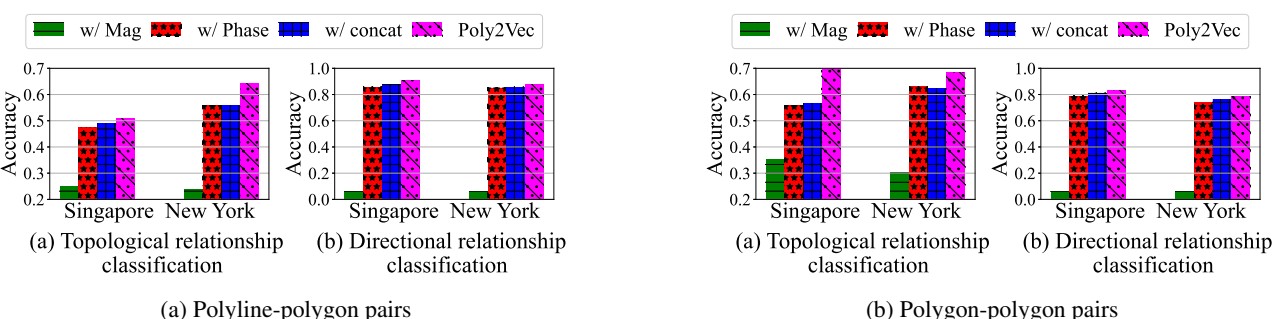

Figure 11: Effect of learned fusion on polyline-polygon and polygon-polygon pairs

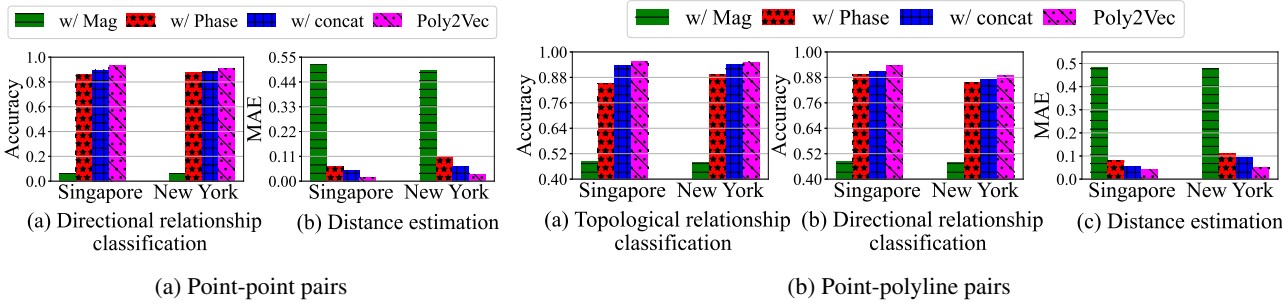

Figure 12: Effect of learned fusion on point-polygon and point-polyline pairs

### A.13.2. EFFECT OF EMBEDDING SIZE $d$

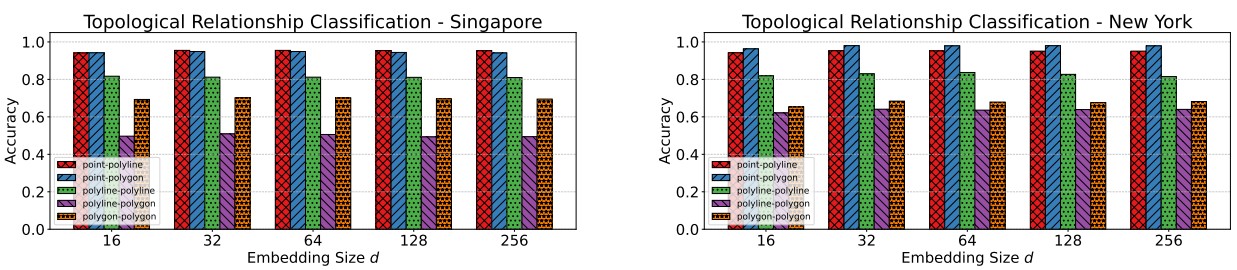

Figure 13: Effect of embedding size on topological relationship classification

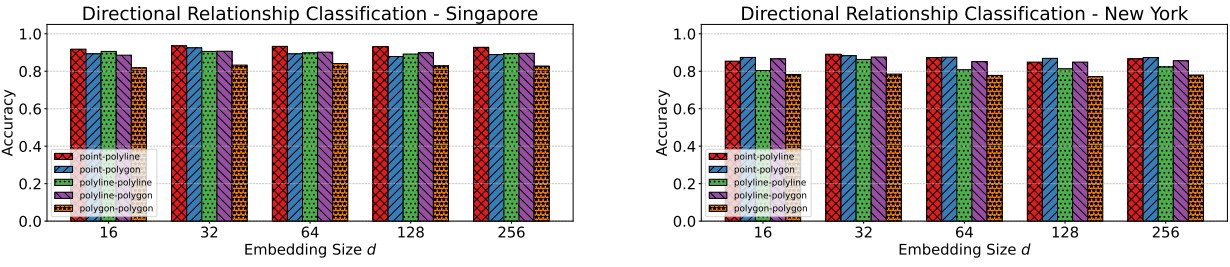

Figure 14: Effect of embedding size on directional relationship classification

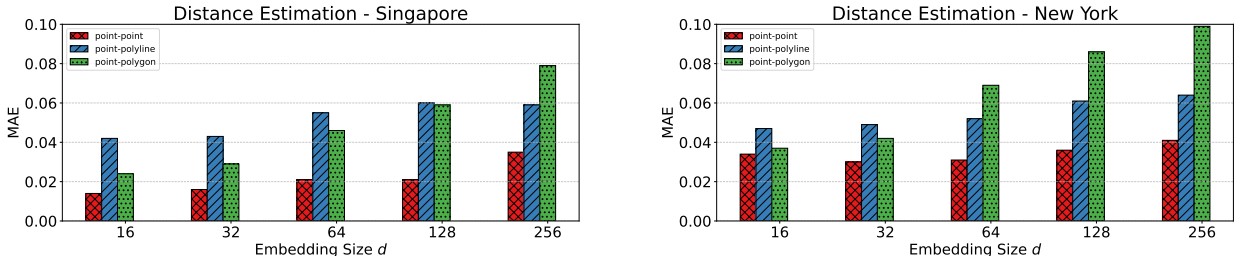

Figure 15: Effect of embedding size in distance estimation

### A.13.3. EFFECT OF NUMBER OF FREQUENCY COMPONENTS $W$

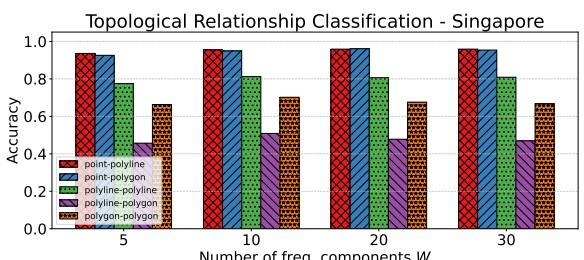
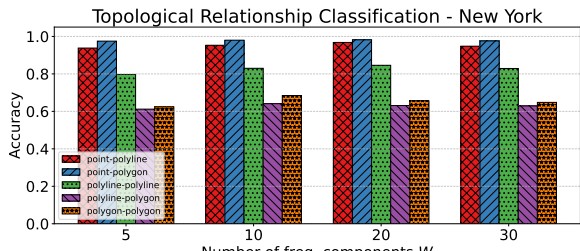

Figure 16: Effect of # of freq. components on topological relationship classification

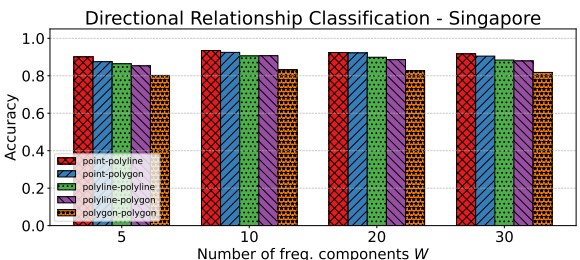
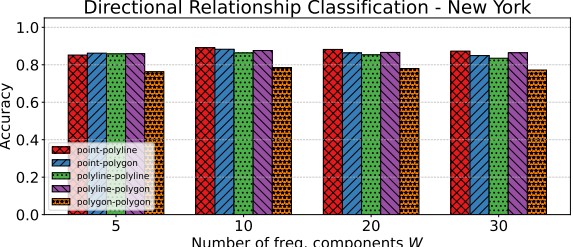

Figure 17: Effect of # of freq. components on directional relationship classification

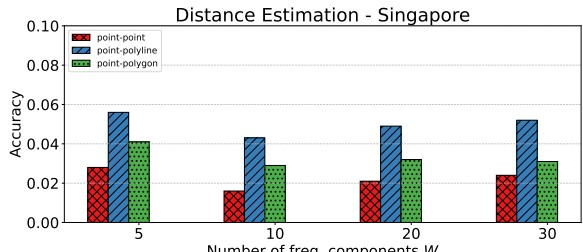
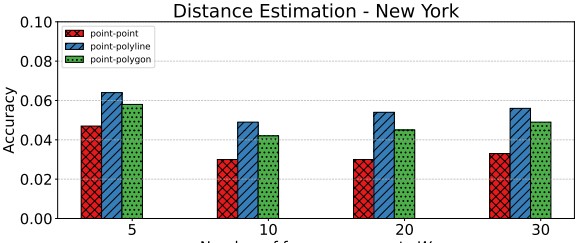

Figure 18: Effect of # of freq. components in distance estimation

