# OpenReview forum: "Poly2Vec: Polymorphic Fourier-Based Encoding of Geospatial Objects for GeoAI Applications"
_ICML.cc/2025/Conference — ICML 2025 poster_

### Official Review · Reviewer_kc8b · 2025-03-12

**Overall Recommendation:** 3

**Summary:**

This paper proposes Poly2Vec, a Fourier-transform approach to encoding shapes (points, lines, polygons) for geospatial tasks. The authors find that Poly2Vec outperforms baselines in preserving shape topology, direction, and distance over OSM datasets for two cities, New York and Singapore.

## Update after rebuttal
I think this paper is good and recommend it for acceptance. I would have liked to see experiments integrating Poly2Vec with a larger geospatial workflow as I think it can benefit geospatial tasks such as super-resolution, conditional generation, and may even benefit unsupervised learning methods for geospatial data. While evidence for these points would have made the paper stronger, in my opinion the contribution as presented by the authors is sufficient to clear the bar for acceptance.

**Claims And Evidence:**

A few key claims and my discussion of the evidence used to support them:

**Poly2Vec outperforms baselines in preserving shape topology, direction, distance:**
I believe evidence is shown to support this claim, as Poly2Vec seems to clearly outperform baselines like Direct, Tile, Wrap, Grid, Theory, and T2Vec. The datasets are limited to two cities, and broader geographic diversity would be welcome though, as I’m guessing the shapes in NYC vs Singapore aren’t too different.

**Integrating Poly2Vec in an end-to-end GeoAI workflow improves performance in population prediction and land-use inference**
This is the more pertinent/interesting claim to me and I think the claim is not strongly supported here. The authors pick a specific baseline, RegionDCL, and show that replacing its distance-biased encodings with Poly2Vec improves the pipeline. This is too narrow of a result to support the broader claim. There are important baselines that use OSM data including through large language models (eg: [1]) that can be used for population prediction. Moreover, a ResNet-18 is too weak of a baseline given that newer, larger foundational models [2,3,4] are used for creating embeddings of locations or images. Does integrating Poly2Vec into these workflows improve the quality of embeddings? I think that would be very valuable to test.

---
References:

[1] Large language models are geographically biased, _ICML 2024_.
[2] Satclip: Global, general-purpose location embeddings with satellite imagery, _AAAI 2025_.
[3] SatMAE: Pre-training Transformers for Temporal and Multi-Spectral Satellite Imagery, _NeurIPS 2022_.
[4] Scale-MAE: A Scale-Aware Masked Autoencoder for Multiscale Geospatial Representation Learning, _ICCV 2023_.

**Essential References Not Discussed:**

See above.

**Experimental Designs Or Analyses:**

I think more ablations are required, e.g., over the number of samples in the frequency domain W, the embedding size, more sampling strategies beyond learned vs geometric etc.

**Methods And Evaluation Criteria:**

The benchmark datasets are a bit too narrow in scope. They do the job of demonstrating that Poly2Vec might outperform other shape-encoding baselines, but the broader claim of relevance to an end-to-end Geospatial workflow is not sufficiently tested over OSM data with two cities.

Moreover, I’m curious to know if linear probing embeddings from satellite image foundation models for distance or topological details of shapes in the image can yield the same accuracies that Poly2Vec displays. This to me is also an important comparison- the authors can use this experiment to demonstrate the value of Poly2Vec’s embeddings in improving the embeddings of geospatial image foundation models.

**Other Comments Or Suggestions:**

N/A

**Other Strengths And Weaknesses:**

Some other strengths:
* Paper is clearly written, easy to follow.
* Improvements over other shape-encoding baselines specifically for topological/directional metrics is clear.

Some other weaknesses:
* Lack of thorough experimental validation (over multiple cities/geographies)
* Insufficient experimental results on more recent geospatial workflows
* Scope of claims is too narrow
* Using Fourier based encodings is not new

I am still slightly leaning towards acceptance because I do think the paper is promising, provided the authors demonstrate empirical evidence over Poly2Vec’s broader utility.

**Questions For Authors:**

See above.

I'm also curious about the ability of Poly2Vec to improve generative image foundation models for geospatial applications (e.g. diffusion), although this is not required to test.

**Relation To Broader Scientific Literature:**

The authors proposed method is promising for geospatial applications, but the key evidence required to demonstrate its utility is through integrations in much larger, more recent geospatial workflows. This is currently lacking in the paper, which instead discusses improvements over other shape encoding baselines (necessary but not sufficient). The authors should spend time surveying more recent geospatial use-cases where poly2vec could demonstrate value, and then share the results of experiments where poly2vec is used to provide embeddings.

**Theoretical Claims:**

N/A

---

> ### Author Rebuttal · Authors · 2025-03-31
>
> Thank you for these detailed, thought-provoking comments.
> ___
> ***Reviewer (paraphrased)***: Existing baselines use OSM data, including LLMs[1]. ResNet-18 is weak given [2,3,4].
>
> ***Authors:*** Poly2Vec is designed to handle arbitrary geometric data described by coordinates (vector spatial data), rather than pixel-based images such as satellite imagery. Thus, methods explicitly requiring imagery are not directly applicable to our scenario.
>
> RegionDCL converts vector data of buildings into images, using ResNet-18 for image encoding and distance-biased transformer for restoring location information of buildings and POIs. We hypothesize that RegionDCL adopted this approach because no effective unified encoding existed for vector data before Poly2Vec. This vector-to-image conversion (rasterization) step inherently introduces information loss. We demonstrate that directly encoding vector data with Poly2Vec is more effective. While using a more sophisticated image-based model might yield incremental improvements, the fundamental loss of spatial detail due to rasterization would persist.
>
> On the suggested references: [1] primarily evaluates large language models, while [3] and [4] focus explicitly on satellite images, and therefore Poly2Vec does not directly apply. Nevertheless, the broader trend in foundation and large language models emphasizes multimodal data integration. [2] exemplifies this integration, combining both images and geographical coordinates; however, its focus appears more oriented toward geographic positioning rather than the locations, topology, or spatial relationships that Poly2Vec exploits. Poly2Vec could be integrated into [2] to replace/complement its location encoder, to support more sophisticated downstream tasks. Similarly, [1], [3], and [4] can benefit from Poly2Vec if they can be extended to incorporate supplementary vector data inputs.
> ___
>
> ***Reviewer (paraphrased)***: Benchmark datasets too narrow in scope. … broader claim of relevance to an end-to-end Geospatial workflow is not sufficiently tested over OSM data with two cities.  Need experiments on larger, more recent workflows.
>
> ***Authors:*** The two cities, New York City and Singapore, have significant differences in building structures, spatial layout, and data density (as visualized in [Link1](https://anonymous.4open.science/r/r-0752/)). New York City has a large, diverse urban environment with varied functional zones and more regular building shapes, while Singapore features a much higher population density, more data-sparse regions, and more irregularly shaped footprints [1]. Poly2Vec focuses on encoding vector data, which serves as the foundation for many geospatial applications. We believe that improving encoding vector data will facilitate a broader range of geospatial tasks.
>
> [1] Urban Region Representation Learning with OpenStreetMap Building Footprints, ACM SIGKDD 2023
>
> ___
> ***Reviewer (paraphrased):*** Linear probing embeddings from satellite image foundation models for distance or topological details of shapes in the image can yield the same accuracies? … demonstrate the value of Poly2Vec’s embeddings in improving the embeddings of geospatial image foundation models.
>
> ***Authors:*** Similar to the first point. On the one hand, satellite image foundation models cannot apply to our settings since we focus on vector data. On the other hand, if a geospatial image foundation model can take vector data as input, so as to inject the spatial information into the embedding, we believe Poly2Vec can be a good choice to encode the coordinates.
> ___
> ***Reviewer (paraphrased):*** More ablations required.
>
> ***Authors:*** We tuned the main hyperparameters before submission. Please find the hyperparameter study in [Link1](https://anonymous.4open.science/r/r-0752/), including an additional sampling technique, namely uniform sampling.
> ____
>
> ***Reviewer (paraphrased):*** Can Poly2Vec  improve generative image foundation models?
>
> ***Authors:*** See 3rd rebuttal response to Reviewer Bw6M.
> ____
> ***Reviewer (paraphrased):*** Using Fourier-based encodings is not new
>
> ***Authors:*** Agreed. Our novel contributions: (1) Leverage Fourier transform to create a unified, polymorphic encoding across different geometry types, with easy extensions to any 2D shape. (2) Fourier transform of line segment is new. Novel polygon approach. (3) Flexibly mixing different geometry types for geometric inferences. (4) Consistently outperforming baselines, including previous frequency-based.
> ____
>
> ***Reviewer (paraphrased):*** Supplementary material input/output of each task can be clearer. Link A.4.1 in main text for clarity.
>
> ***Authors:*** For the sake of reproducibility, we provide the details of all our experimental settings. But given the space limitation, we can only present them in the appendix. To improve clarity and accessibility, we will link to each appendix subsection in the main paper and also add backward references in the appendix.
> ____

---

> > ### Comment · Reviewer_kc8b · 2025-04-02
> >
> > I thank the authors for their rebuttal response. Some of my points have been clarified.
> >
> > However, I still think it would be valuable to demonstrate additional experimental validation of Poly2Vec in a geospatial workflow beyond RegionDCL.
> >
> > * I think it could be quite interesting to see whether complementing the SatCLIP [2] location encoder with Poly2Vec improves the quality of its representations.
> > * For SatMAE [3], I think augmenting the "temporal" or "multi-spectral" encodings with Poly2Vec encodings, and then either pre-training or fine-tuning with these augmentations would be a valuable demonstration of Poly2Vec's utility.
> > * For a work like DiffusionSat, it would be quite interesting to see if the quality of generated images can be improved with Poly2Vec encodings passed to the metadata encoder, or to a conditioning ControlNet.
> >
> > There are likely other examples as well of demonstrating Poly2Vec's broader utility in a larger geospatial workflow, the above are suggestions. The current manuscript does a good job of demonstrating Poly2Vec as a solid improvement over prior shape encoding baselines. For me to increase my score, I would like to see more evidence of Poly2Vec's utility in larger geospatial workflows like the ones suggested above.

---

> > > ### Author Response · Authors · 2025-04-05
> > >
> > > We appreciate the reviewer’s suggestion to include additional content. However, we would like to note that the paper already contains a substantial core contribution that required significant space to develop both theoretically and empirically.
> > >
> > > The central innovation of our work lies in the novel application of the Fourier transform to spatial data, enabled by defining indicator functions over basic geometric shapes, specifically line segments and triangles, for which closed-form Fourier representations can be derived. To the best of our knowledge, this formulation has not been explored previously. Building on this foundation, we show how complex yet commonly used spatial objects, such as lines (modeled as ridges of delta functions) and polygons (decomposed into triangles), can also be represented within this framework. Through a careful combination of these shape formulations and affine transformations, we develop a unified and expressive vector representation for points, polylines, and polygons that captures their geometry, spatial location, and inter-topological relationships.
> > >
> > > This pipeline is, in our view, a significant step forward. It required us to carefully formulate the theory, justify it mathematically, and validate it through extensive experiments. These components together occupied much of the available space in the paper, leaving limited room to add further material without compromising clarity or focus.
> > > With this in mind, we would like to clarify that the examples suggested by the reviewer, while referred to as part of a “geospatial workflow”, do not involve spatial objects such as points, polylines, or polygons as inputs. Instead, they primarily use location-based, temporal, or multi-spectral data, which are beyond the scope of this paper. As such, we do not believe these pipelines are suitable for demonstrating the effectiveness of Poly2Vec, despite the terminological overlap. The term “geospatial” can indeed be overloaded, and our focus here is specifically on the representation and reasoning over geometric spatial objects.

---

### Official Review · Reviewer_qMT2 · 2025-03-12

**Overall Recommendation:** 4

**Summary:**

The authors introduce a method of encoding representations of geospatial objects which they call Poly2Vec. This method is capable of representing points (e.g. points of interest), polylines (e.g. roads), and polygons (e.g. buildings) while competing methods struggle to represent all of these different formats. Poly2Vec is a fourier transform based approach which seems to preserve some important attributes of the encoded objects such as location, shape and direction. This method is compared to a wide range of other approaches on OpenStreetMap based datasets for tasks such as determining the distance between objects or their orientation, as well as for use as input encodings for a modern machine learning approach to land use classification and population prediction. Poly2Vec performs best on all of these tasks.

## update after rebuttal

The author response clarified my one concern and I believe the paper should be accepted.

**Claims And Evidence:**

Yes. The authors seem to compare to a wide variety of competing approaches and show favourable performance across a range of tasks and appropriate metrics.

**Essential References Not Discussed:**

Essential references seem to be included.

**Experimental Designs Or Analyses:**

All experiments seem to be well designed.

**Methods And Evaluation Criteria:**

Yes. The method seems sensible. The OpenStreetMap based datasets seem sensible for the task and similar datasets are used in the literature (“Urban Region Representation Learning with OpenStreetMap Building Footprints” (KDD '23), “Urban2Vec: Incorporating Street View Imagery and POIs for Multi-Modal Urban Neighborhood Embedding” (AAAI '20)). The evaluation metrics seem appropriate. Ground truth data for tasks such as land use classification has been gathered similarly to previous works in the area.

**Other Comments Or Suggestions:**

Well put together paper and not many mistakes that I can see, but on line 875 we have a couple of “Figure ??”s.

**Other Strengths And Weaknesses:**

The work seems to be quite original in furthering spectral based methods of encoding polygons to also include points and lines.

The paper is clear and the work done is well described. Baseline approaches are covered quite well.

The tasks seem well constructed and a wide range of approaches are compared against.

Several different evaluation metrics are used and the proposed method is consistently well performing.

The method involves learning how to encode objects in a way that seems to allow it to be fine tuned to specific tasks which might not be helpful for a downstream user who does not wish to train a model in order to encode their geospatial objects, and might limit the transferability of these representations. I also wonder if including a similar neural network for some of the baseline approaches and training these to produce encodings that are useful for specific tasks would also improve performance of these baselines and make Poly2Vec comparatively less high performing.

**Questions For Authors:**

1. The method involves learning how to encode objects in a way that seems to allow it to be fine tuned to specific tasks which might not be helpful for a downstream user who does not wish to train a model in order to encode their geospatial objects, and might limit the transferability of these representations. Do the authors know if including a similar neural network for some of the baseline approaches and training these to produce encodings that are useful for specific tasks would also improve performance of these baselines and make Poly2Vec comparatively less high performing? This will allow me to better judge the performance of the proposed method.

**Relation To Broader Scientific Literature:**

Some existing works involve creating representations of buildings or places of interest but these make use of semantic information such as text or photos of the location. “Urban Region Representation Learning with OpenStreetMap Building Footprints” (KDD '23)

Many works involve generating an embedding of points and some of these are compared against in this work (e.g. WRAP method in "Presence-Only Geographical Priors for Fine-Grained Image Classification" (ICCV '19)). These can be extended to represent lines as sequences of points. (e.g. "LSTM-TrajGAN: A Deep Learning Approach to Trajectory Privacy Protection" (GIScience 2021))

“Towards General-Purpose Representation Learning of Polygonal Geometries” (GeoInformatica 27(4):1-52) produces encodings of polygons that are suitable for machine learning approaches.

“Graph Convolutional AutoEncoder models” can also produce encodings of polygons or lines.

Spectral domain polygon encoders do exist such as “NUFTSPEC”. The authors do cite this work. (“Towards General-Purpose Representation Learning of Polygonal Geometries” (GeoInformatica 27(4):1-52)).

Poly2Vec does seem to uniquely create a unified embedding space for the different geometries that are discussed in the work (polygons, polylines, points)

Deterministic spatial reasoners such as postGIS can do many of the tasks that are discussed in this paper such as accurately measuring distances between polygons, lines and points, and determining directions. However these do not produce an encoding that is suitable for many machine learning approaches.

**Theoretical Claims:**

I couldn’t see any problems with the formulation of the fourier transform based method.

---

> ### Author Rebuttal · Authors · 2025-03-31
>
> Thank you for your insightful comments.
>
> ___
>
> ***Reviewer:*** (Note we grouped these comments, because our answer is essentially the same for all of them.) The method involves learning how to encode objects in a way that seems to allow it to be fine tuned to specific tasks which might not be helpful for a downstream user who does not wish to train a model in order to encode their geospatial objects, and might limit the transferability of these representations. … I also wonder if including a similar neural network for some of the baseline approaches and training these to produce encodings that are useful for specific tasks would also improve performance of these baselines and make Poly2Vec comparatively less high performing. … The method involves learning how to encode objects in a way that seems to allow it to be fine tuned to specific tasks which might not be helpful for a downstream user who does not wish to train a model in order to encode their geospatial objects, and might limit the transferability of these representations. Do the authors know if including a similar neural network for some of the baseline approaches and training these to produce encodings that are useful for specific tasks would also improve performance of these baselines and make Poly2Vec comparatively less high performing? This will allow me to better judge the performance of the proposed method.
>
> ***Authors:*** We understand this question and appreciate the trigger to think about this more deeply. You are right that we train our MLPs specifically for the task. Please note that we do the same for the baselines, so our performance comparisons are fair. We will clarify this in the next version of the paper. As we explain in the paper (Section 3.2), different parts of the Fourier transform are important for different tasks. E.g. the Fourier transform’s magnitude generally encodes shape, and its phase generally encodes location. Thus the targeted training tends to extract the right information for the specialized task.  Learning general-purpose embeddings is also an interesting direction that we have considered. Such representations could be learned using unsupervised or self-supervised techniques, such as encoder-decoder frameworks and contrastive learning.
>
> ___
>
> ***Reviewer:*** Well put together paper and not many mistakes that I can see, but on line 875 we have a couple of “Figure ??”s.
>
> ***Authors:*** Thank you for finding these omissions. We will fix them all in the next version.

---

> > ### Comment · Reviewer_qMT2 · 2025-04-04
> >
> > Thank you for your helpful response.
> >
> > Just to clarify, for a baseline approach such as "Wrap", after the trigonometric encoding of the input coordinates, you include an MLP similar to the "learned fusion" module to allow these relatively simple representations to be adapted to the downstream task?

---

> > > ### Author Response · Authors · 2025-04-05
> > >
> > > Thank you for the follow-up question. The paper includes two MLPs. The first is part of our learned fusion module, which is specific to Poly2Vec’s design and is not applied to any baselines. The second MLP is a task adaptor that maps encodings for the downstream tasks. We used this second 2-layer MLP adaptor for both Poly2Vec and all the baselines to ensure fair comparison. Our ablation studies (Section 4.3) show that even the Poly2Vec variant that concatenates phase and magnitude (without the learned fusion MLP, referenced as "w/concat") still outperforms all baselines and matches Theory on the polygon-point distance estimation task.

---

### Official Review · Reviewer_Bw6M · 2025-03-14

**Overall Recommendation:** 5

**Summary:**

The paper presents a Fourier based encoding strategy for geospatical principles, i.e., points, lines, and polygons, into a deep-learning compatible vector format. The methodology is well-explained and founded in signal processing theory. It well-explains Fourier analysis and inherent properties (Affine Transformation, Symmetry) that are used in a effective way to first (affine) transform geospatial primitives and then encode them into frequency domain to a vector representations for amplitude and phase. Then two MLPs fuse these representations to map the features vector to a target variable, for instance, for point-in-polygon testing.

Overall, it is well-written. The principles are well-explained and the results well structured into underlying research questions and numerically convincing, as the proposed Poly2vec representation outperforms existing point-based and line-based methods. In my opinion, a strong accept.

**Claims And Evidence:**

The paper claims to unify representation of geospatial objects, namely points, lines, and polygons. This claim is supported, as the results show better representation accuracy over point- and line-specific models.

**Essential References Not Discussed:**

I think the references are all justified. No additions requested.

**Experimental Designs Or Analyses:**

The experiments are well-structured according to 4 research questions. However, all results seem to be based on the classification of point-to-polyline etc relationships, which is reasonable for this experimental setup, but a more realistic usecase, like predicting, for instance, the building use (residential, factory, shopping mall) from the polygion geometry may be interesting as well. For classic geospatial operations like point-in-polygon prediction highly optimized algorithms exist that achieve 100% accuracy (<- but this is still a good problem to compare different embeddings)

**Methods And Evaluation Criteria:**

Methods:
The underlying method is well-explained first on simple numerical examples that are then abstracted to the generic case. Overall, very intuitive and understandable

Evaluation
The experiments are well-structured in 4 accurate reserach questions that are systematically investigated in the results.

**Other Comments Or Suggestions:**

no typos found.

**Other Strengths And Weaknesses:**

Strengths
* Convincing results against baselines focusing on a singly type of geometric primitive
* well-explained methodology and well-structured results

Weakness:
* some more realistic datasets could be tested. E.g., classification of building type, or land use from the geometry shape

**Questions For Authors:**

* Can the Fourier transformation be inverted to map back from embedding space to polygon/line/points?
* What usecases do the authors see for Poly2Vec in a more complex setting, where the fourier features are integrated in a deeper neural network rather than individual MLPs for point-in-polygon tests.

**Relation To Broader Scientific Literature:**

This paper presents a crucial step forward embedding geospatial primitives, which is highly helpful in learning from geospatial data in Geospatial Foundation Models. The unifying framework that includes points, lines, polygons and outperforms methodologies designed for single primitives like points, will be important for the geospatial machine learning community and subsequent research fields like Geo-Information Science.

**Theoretical Claims:**

There are no proofs in the paper, but properties of Fourier transform (Linearity, Affine Trasnformation, Hermitian Symmetry, Magnitude and Phase) are well-explained. All used theory is supported by intuition and concrete experiments on simple cases.

---

> ### Author Rebuttal · Authors · 2025-03-31
>
> Thank you for the thorough review and encouraging comments.
>
> ___
>
> ***Reviewer:*** The experiments are well-structured according to 4 research questions. However, all results seem to be based on the classification of point-to-polyline etc relationships, which is reasonable for this experimental setup, but a more realistic use case, like predicting, for instance, the building use (residential, factory, shopping mall) from the polygon geometry may be interesting as well.
>
> ***Authors:*** While we do not perform single-building classification, our RegionDCL experiments address a more general version of that task: predicting region-level attributes from groups of buildings. This can be seen as a superset of building classification, as it involves not just the shape of a single polygon, but also the locations and inter-relationships of multiple buildings, which is particularly beneficial for our downstream tasks where spatial relationships between objects play a critical role.
>
> ___
>
> ***Reviewer:*** some more realistic datasets could be tested. E.g., classification of building type, or land use from the geometry shape
>
> ***Authors:*** Same as our first response above.
>
> ___
>
> ***Reviewer:*** Can the Fourier transformation be inverted to map back from embedding space to polygon/line/points?
>
> ***Authors:*** We thank the reviewer for the thought-provoking question. We believe this can be achieved by simultaneously training a decoder to map embeddings back to geometries (i.e., shape decoder). So far, we have experimented with applying the inverse Fourier transform to reconstruct approximations of the original shapes from the raw Fourier samples. This is an exciting direction and we will consider it in future research toward generative models that can reconstruct realistic spatial geometries from learned embeddings.
>
> ___
>
> ***Reviewer:*** What use cases do the authors see for Poly2Vec in a more complex setting, where the fourier features are integrated in a deeper neural network rather than individual MLPs for point-in-polygon tests.
>
> ***Authors:*** Our experiments with RegionDCL show one example of integrating Poly2Vec into a transformer-based geospatial pipeline, by replacing its input representation with Poly2Vec to classify regions and estimate population from geometric map features. Looking forward, we believe Poly2Vec can serve as a valuable component in multimodal geo-foundation models, when one of the input modalities is vector-based spatial data (i.e., coordinates).

---

### Official Review · Reviewer_wSi2 · 2025-03-17

**Overall Recommendation:** 3

**Summary:**

This work considers the problem of encoding points, polylines, and polygons for the purposes of prediction tasks that require understanding of spatial relationships such as topology, direction, and distance. Similar to previous point and polygon encoding approaches (Space2vec, NUFTspec), Fourier transformation is used to transform vector space data to fixed length presentations. To achieve more detailed representation of alignment both magnitude and phase formation are encoded. To achieve encoding of mixed item types (point, polyline, polygon) three versions of Fourier transformations are used together.

For basic spatial relationship tasks (topology, direction, distance), experiment shows improvement over previous approaches such as Space2vec, ResNet1D, T2Vec and NUFTspec.

For mixed item type tasks (Land use classification, population prediction), an unsupervised  approach RegionDCL is used as the baseline. Improvement of RegionDCL performance is achieved by replacing its distance based attention bias with the proposed Poly2Vec approach. However, not much detail is given in the paper about how exactly is Poly2Vec integrated to RegionDCL. It would also be useful to provide visualization of a few examples to help the reader understand how Poly2Vec helps RegionDCL.

Overall the proposed approach is reasonable, but there is significant detail missing which makes it hard for me to understand the result.

**Claims And Evidence:**

see summary

**Essential References Not Discussed:**

no

**Experimental Designs Or Analyses:**

see summary

**Methods And Evaluation Criteria:**

see summary

**Other Comments Or Suggestions:**

see summary

**Other Strengths And Weaknesses:**

see summary

**Questions For Authors:**

see summary

**Relation To Broader Scientific Literature:**

No

**Theoretical Claims:**

see summary

---

> ### Author Rebuttal · Authors · 2025-03-31
>
> We thank the reviewer for the insightful comments.
> ___
>
> ***Reviewer:*** Similar to previous point and polygon encoding approaches (Space2vec, NUFTspec), Fourier transformation is used to transform vector space data to fixed length presentations.
>
> ***Authors:*** We note that our Poly2Vec differs from Space2Vec in that (1) Poly2Vec has the full complement of frequency components available to it, as opposed to Space2Vec’s more tightly specified sine and cosine frequencies on a grid of scales and (2) Poly2Vec works uniformly for points, polylines and polygons, while Space2Vec is limited to points. Compared to NUFTspec, Poly2Vec presents a unified, consistent Fourier transform encoding for points and polylines, as well as polygons. Our approach for computing the Fourier Transform for polylines and polygons (through decomposition and affine transformation) is also a novel component of our approach.
> ___
> ***Reviewer:*** Improvement of RegionDCL performance is achieved by replacing its distance based attention bias with the proposed Poly2Vec approach. However, not much detail is given in the paper about how exactly is Poly2Vec integrated to RegionDCL.
>
> ***Authors:*** We agree that clearly describing Poly2Vec’s integration into RegionDCL would improve the manuscript’s clarity. Due to space constraints and since RegionDCL itself was not our primary contribution, we focused instead on detailing our novel components.
>
> To clarify, Poly2Vec is used as the input encoding in RegionDCL, replacing its original input representation. RegionDCL originally rasterizes OSM building footprints, converting coordinate data into image inputs so it can leverage convolutional encoders like ResNet-18. This rasterization leads to the loss of important spatial information, such as the absolute location of each building. To mitigate this, RegionDCL introduces a distance-biased transformer encoder, where the bias term consists of pairwise distances between buildings and POIs to reintroduce spatial context. In our experiments, we (1) replaced the inputs with Poly2Vec encodings, and (2) replaced the distance-biased transformer encoder with a standard transformer encoder,  because our new inputs from (1) capture the necessary spatial information. The fact that Poly2Vec improves performance even without the distance bias demonstrates its ability to inherently retain spatial and positional information.
>
> We will add these additional details in the appendix to clearly describe the integration and highlight the differences from the original setup of RegionDCL.
> ___
>
> ***Reviewer:*** It would also be useful to provide visualization of a few examples to help the reader understand how Poly2Vec helps RegionDCL.
>
> ***Authors:*** We provide some visualization examples in [Link1](https://anonymous.4open.science/r/r-0752/), showing building footprints as polygons and their spatial relationships — information that, discarded by the original design of RegionDCL but captured by Poly2Vec, contributes to improved classification performance in RegionDCL. We will add such visualizations of selected regions in the new manuscript.

---

> > ### Comment · Reviewer_wSi2 · 2025-04-04
> >
> > Thanks for the added explanations. I have raised my score by 1.
> > However, I still think the paper can benefit from visualizing the representation somehow in order to reveal how does it work to solve the geographical problems.

---

> > > ### Author Response · Authors · 2025-04-09
> > >
> > > Thank you for raising your score. We appreciate your feedback, and we will consider it carefully as we revise this paper and future explanations.

---

### Decision · Program_Chairs · 2025-05-01

**Decision:**

Accept (poster)

**Comment:**

After the discussion phase, all reviewers recommended acceptance (Strong Accept, Accept, 2x Weak Accept) noting that the paper is well written, clearly motivated, has novelty, and is sufficiently evaluated. The rebuttal addressed many of the reviewer concerns, such as clarifying details of the baselines. As a result, the AC decided to accept the paper. Please take the reviewer feedback into account when preparing the camera-ready version.